# Mapping *Plasmodium* transitions and interactions in the *Anopheles* female

Yan Yan[1,9], Lisa H. Verzier[1,9], Elaine Cheung[1,9], Federico Appetecchia[1], Sandra March[2,3,4,5], Ailsa R. Craven[1], Esrah Du[1], Alexandra S. Probst[1], Tasneem A. Rinvee[1], Laura E. de Vries[1], Jamie Kauffman[1], Sangeeta N. Bhatia[2,3,4,5,6], Elisabeth Nelson[1], Naresh Singh[1], Duo Peng[1,7], W. Robert Shaw[1,8] & Flaminia Catteruccia[1,8 ✉]

The human malaria parasite, *Plasmodium falciparum*, relies exclusively on *Anopheles* mosquitoes for transmission. Once ingested during blood feeding, most parasites die in the mosquito midgut lumen or during epithelium traversal[1]. How surviving ookinetes interact with midgut cells and form oocysts remains poorly understood, yet these steps are essential to initiate a remarkable growth process culminating in the production of thousands of infectious sporozoites[2]. Here, using single-cell RNA sequencing of both parasites and mosquito cells across different developmental stages and metabolic conditions, we unveil key transitions and mosquito–parasite interactions that occur in the midgut. Functional analyses uncover processes that regulate oocyst growth and identify the *Plasmodium* transcription factor PfSIP2 as essential for sporozoite infection of human hepatocytes. Combining shared mosquito–parasite barcode analysis with confocal microscopy, we reveal that parasites preferentially interact with midgut progenitor cells during epithelial crossing, potentially using their basal location as an exit landmark. Additionally, we show tight connections between extracellular late oocysts and surrounding muscle cells that may ensure parasite adherence to the midgut. We confirm our major findings in several mosquito–parasite combinations, including field-derived parasites. Our study provides fundamental insight into the molecular events that characterize previously inaccessible biological transitions and mosquito–parasite interactions, and identifies candidates for transmission-blocking strategies.

The apicomplexan parasite *P. falciparum* is responsible for 90% of cases of malaria, a disease that in 2023 alone caused the death of close to 600,000 people, mostly young children in sub-Saharan Africa[3]. Like all malaria parasites, *P. falciparum* requires a blood-feeding vector for transmission between humans, and over the course of a long co-evolutionary history it has adapted to be exclusively transmitted by mosquitoes of the *Anopheles* genus. The parasite journey in the *Anopheles* female starts when male and female gametocytes are ingested during a blood meal and rapidly mature into gametes, leading to fertilization and formation of a zygote which transforms into a motile ookinete in the blood bolus. Between 24 h and 36 h after ingestion, ookinetes must cross the midgut epithelium to avoid being killed during blood digestion. These initial steps result in substantial parasite loss, and represent one of the most severe bottlenecks in the parasite life cycle[1]. After reaching the basal side of the midgut, ookinetes round up via an intermediate 'took' (transforming ookinete) stage[4] to transform into extracellular oocysts, which over the next 7–10 days will extensively grow in size just beneath the basal lamina surrounding the midgut epithelial layer while undergoing DNA replication. This remarkable growth process culminates in the formation of thousands of infectious sporozoites, an intricate process in which individual parasites must be segmented precisely over a relatively short period of time[2] (Fig. 1a). When mature, sporozoites egress from oocysts and invade the mosquito salivary glands, from where they can be transmitted to the next person when the mosquito bites again.

The mosquito stages of *P. falciparum* development are relatively poorly characterized compared with asexual blood stages, despite their essential role in transmission, and current knowledge is largely extrapolated from work on more tractable rodent malaria models[5–10]. Although several studies have visualized the route by which ookinetes invade the midgut epithelium[11–15], whether they interact with specific midgut cell types and the mechanisms that govern their transition into oocysts remain largely unknown. The processes that fuel oocyst development and promote sporozoite formation are also not fully understood. Tackling these questions has been challenging, as mosquito stages of *P. falciparum* are restricted to in vivo study and have limited genetic tools relative to asexual blood stage parasites[16]. Moreover, the low ookinete and oocyst load in a midgut limits genome-wide multi-omics approaches of those stages, which have therefore been mostly confined to non-human *Plasmodium* species, where parasite

[1]Department of Immunology and Infectious Disease, Harvard T. H. Chan School of Public Health, Boston, MA, USA. [2]Howard Hughes Medical Institute, Cambridge, MA, USA. [3]Institute for Medical Engineering and Science, Massachusetts Institute of Technology (MIT), Cambridge, MA, USA. [4]David H. Koch Institute for Integrative Cancer Research, MIT, Cambridge, MA, USA. [5]Broad Institute of MIT and Harvard, Cambridge, MA, USA. [6]Wyss Institute at Harvard University, Boston, MA, USA. [7]The Chan Zuckerberg Biohub, San Francisco, CA, USA. [8]Howard Hughes Medical Institute, Boston, MA, USA. [9]These authors contributed equally: Yan Yan, Lisa H. Verzier, Elaine Cheung. ✉e-mail: fcatter@hsph.harvard.edu

densities are substantially higher[17–20]. For similar reasons, single-cell RNA-sequencing (scRNA-seq) studies of *P. falciparum* in the mosquito have focused on stages during which parasites are numerous and readily isolated (that is, blood bolus stages and sporozoites)[21,22], leaving significant gaps in the parasite life cycle.

Here we reconstruct critical bottleneck stages of parasite growth in the midgut and identify mosquito–parasite interactions by performing dual scRNA-seq of parasites and midgut cells over time, capturing key processes that mediate midgut crossing, oocyst growth and the initial stages of sporozoite formation.

## scRNA-seq of midgut and parasite cells

We infected *Anopheles gambiae*–the major malaria vector in sub-Saharan Africa–with *P. falciparum* and isolated both parasites and midgut cells at 4 different time points: invading ookinetes (36 hours post-infection (hpi)); newly formed oocysts (2 days post-infection (dpi)); growing oocysts (4 dpi); and late oocysts that may have begun sporozoite segmentation (7 dpi) (Fig. 1a and Extended Data Fig. 1a). We did not attempt to isolate oocysts at later time points given their large size (more than 50 μm), which exceeds the limits of 10x technology. Owing to low parasite numbers relative to the overwhelming number of mosquito cells[23], we enriched for parasites by optimizing the single-cell isolation protocol (Methods) and applied deeper sequencing. In brief, infected midguts were partially digested with collagenase IV and elastase, then filtered through a series of cell strainers to remove large clumps of mosquito cells. The resulting single-cell suspensions contained both midgut cells and parasites, with a cell viability of over 93% as determined by trypan blue staining. We compared parasites developing in mosquitoes under two metabolic conditions: control mosquitoes (injected with double-stranded RNA targeting *eGFP* (ds*GFP*)) and mosquitoes depleted of the ecdysone receptor (ds*EcR*), the nuclear co-receptor of the steroid hormone 20-hydroxyecdysone[24] (Extended Data Fig. 1b–e). Impairing 20-hydroxyecdysone signalling reliably accelerated oocyst growth, which we reasoned would provide more granular resolution at the later stages, and did not affect parasites numbers as previously observed at low parasite densities[24,25] (Extended Data Fig. 1b,c). After removing low-quality parasites with low transcript and gene counts and high mitochondrial percentage, we successfully profiled 3,495 parasites, detecting a median of 242–1,773 genes at the various time points and treatments (Extended Data Fig. 1f and Supplementary Table 1).

Principal component analysis (PCA) revealed a large separation along the first component, with parasites at 2, 4 and 7 dpi clustering away from 36 hpi parasites (Fig. 1b). The ookinete–oocyst transition (2 dpi versus 36 hpi) was characterized by a downregulation of RNA metabolism paralleled by an upregulation of translation, and oocyst growth was marked by increased aerobic respiration (4 dpi versus 2 dpi) (Extended Data Fig. 2 and Supplementary Table 2). PCA at early time points showed no clear separation between parasites developing in ds*GFP* and ds*EcR* mosquitoes, and consistently, there was no difference in gene expression between these groups. At 7 dpi, however, we detected the upregulation of several genes in parasites from ds*EcR* female mosquitoes (Extended Data Fig. 2 and Supplementary Table 2), probably reflecting the more advanced oocyst stage in this group (Extended Data Fig. 1b). Among the upregulated genes were the gene encoding circumsporozoite protein (CSP; the most abundant sporozoite surface protein) and genes from the rhoptries, secretory organelles that are essential for invasion of host cells[26], suggesting that we successfully captured the start of sporozoite segmentation.

## Parasite atlas across all midgut stages

Next, we generated a uniform manifold approximation and projection (UMAP) of parasites in the mosquito midgut by integrating our datasets with parasites from previous scRNA-seq studies[21,22], including gametes and blood bolus ookinetes as well as sporozoites isolated from oocysts (Fig. 1c). This complete midgut atlas uncovered 11 parasite clusters, which we annotated on the basis of marker genes and data source as follows: gametes/zygotes, blood bolus ookinetes, invading ookinetes (Ookinete 1 and 2), Took, newly formed oocysts (day 2 oocyst), growing oocysts (Oocyst 1–4) and oocysts that are segmenting into sporozoites (Oocyst spz) (Fig. 1d,e and Extended Data Fig. 3a, Supplementary Table 3). In agreement with our observation of early segmentation signals at 7 dpi, a few cells bridged the gap between oocysts in an advanced growth phase and sporozoites isolated from oocysts (Fig. 1c,d and Extended Data Fig. 3b). Pseudotime analysis revealed a single trajectory aligning with the expected clusters (Fig. 1f).

## Gene clusters unveil key transitions

We clustered genes with similar expression patterns over pseudotime, using only our data owing to differences in gene detection rates among scRNA-seq techniques. The 15 gene clusters identified in this analysis separated into 4 major branches (Fig. 1g and Supplementary Table 4). The first branch (gene clusters 6 and 9) captured expected differences between blood bolus ookinetes and ookinetes that are crossing the epithelium. Cluster 6, highly expressed in blood bolus parasites, was associated with ookinete maturation and characterized by genes involved in cytoskeleton and myosin complex[27]. Cluster 9, expressed across all ookinete clusters, comprised micronemal proteins that are critical for motility and midgut traversal, such as circumsporozoite- and TRAP-related protein (CTRP), chitinase and glycosylphosphatidylinositol-anchored micronemal antigen (GAMA)[28,29].

The second branch was largely characterized by processes related to transcription, surface remodelling and adhesion–notable, given that this branch encompasses a transition (ookinete–oocyst) during which parasites undergo profound morphological changes. At the onset of this transition (clusters 13 and 2) we detected strong signals of genes encoding proteins involved in helicase activity and RNA processing, including splicing. Clusters 2 and 5 instead comprised membrane and vesicle trafficking genes, which may facilitate protein transport and surface remodelling. Additionally, two clusters in this branch (13 and 11) contained several *var* genes encoding variants of PfEMP1 proteins, which are well characterized for their adhesion properties in the asexual blood stage[30]. We speculate that young oocysts may use PfEMP1 to anchor themselves between midgut cells and basal lamina, similar to the manner in which asexual parasites adhere to the lining of blood vessels[30].

The largest branch, spanning from day 2 oocyst to Oocyst spz, captured the processes fuelling the growth phase, including genes regulating translation, proteasome activity and ribonucleoprotein complex (clusters 12, 8 and 7) as well as genes involved in aerobic respiration, lipid biosynthesis and apicoplast functions (clusters 8, 1 and 4). Gene cluster 4, highest in oocyst 4, included functions related to the cell cycle, such as DNA replication (Fig. 1g and Supplementary Table 4).

The final branch, highly expressed in Oocyst spz, was characterized by clear signatures of sporozoite segmentation. Clusters 3, 10, 14 and 15 included genes involved in nuclear migration, components of the inner membrane complex (a scaffolding compartment essential for daughter cell formation) and the cytoskeleton network supporting it, and genes from the invasive organelles rhoptries and micronemes–probably preparing sporozoites for salivary gland and hepatocyte invasion[26,31,32] (Fig. 1g and Supplementary Table 4).

Although poly-A capture is not designed to target ribosomal RNAs (rRNAs), these molecules are ubiquitously present in *Plasmodium* bulk and single-cell RNA-sequencing data[33]. We detected limited expression of different rRNAs via random binding, organized in distinct gene expression patterns (Extended Data Fig. 3c). Expression of the A1/A2 forms that are dominant in asexual parasites[34] appeared highest in ookinete clusters, then gradually decreased–although not entirely to

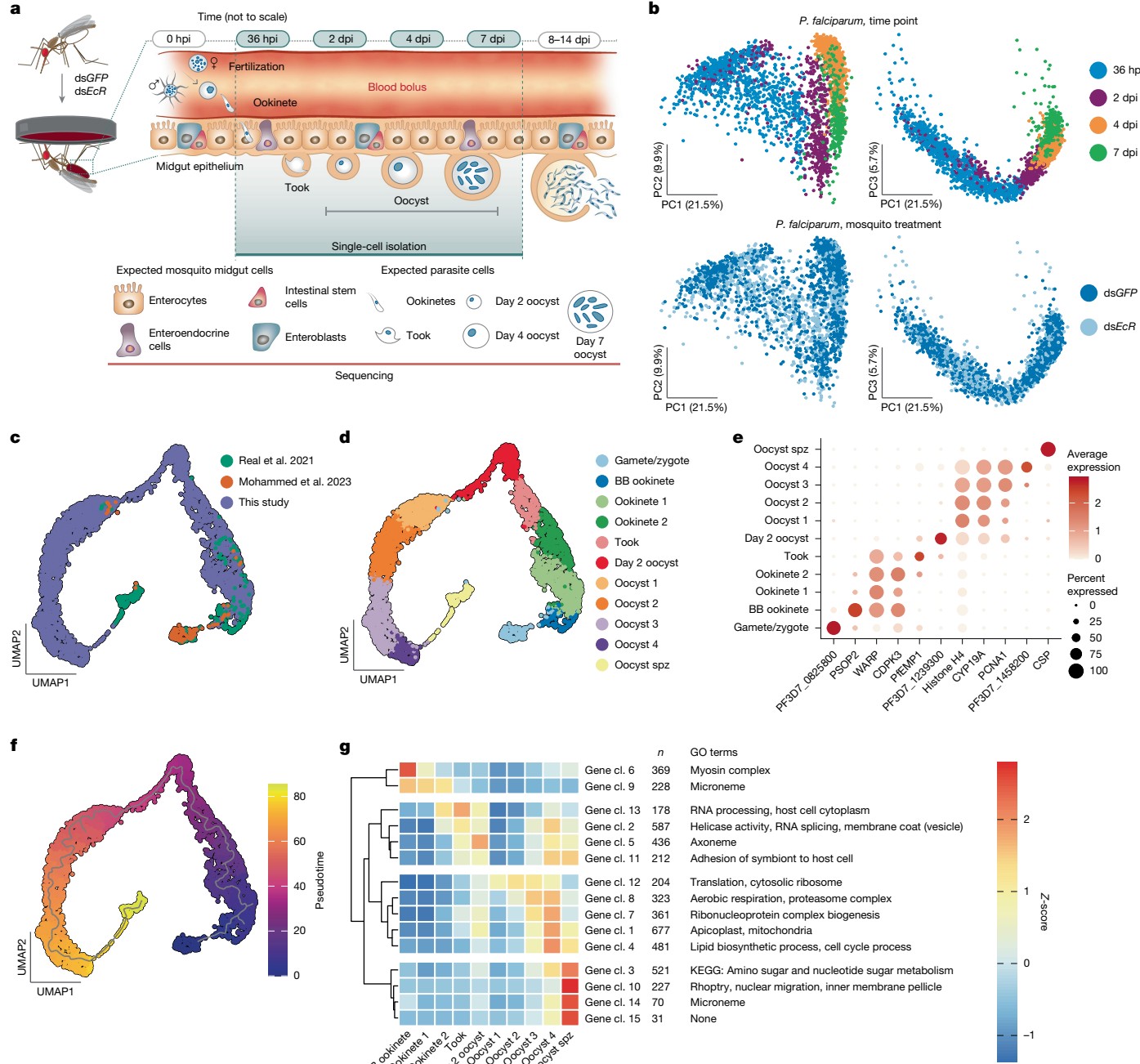

**Fig. 1 | scRNA-seq provides a complete map of *P. falciparum* midgut stages.**
**a**, Schematic of dual scRNA-seq, displaying expected parasite forms and mosquito midgut cell types. Single *P. falciparum* parasites and *A. gambiae* midgut cells were collected from 80 GFP-depleted (control) and 80 EcR-depleted mosquitoes at the time points indicated by the shaded area: 36 hpi, 2 dpi, 4 dpi and 7 dpi and across four biological replicates. **b**, PCA plots of the 3,495 parasites collected across the 4 time points (top) and the 2 treatments (bottom). **c**,**d**, UMAP plots integrating our scRNA-seq data with midgut stage data from two other studies[21,22] (**c**) and showing 11 distinct parasite clusters (**d**). BB, blood bolus. **e**, Dot plot of the top marker genes for each of the parasite clusters, showing their normalized average expression (colour) and the proportion of parasites expressing each gene (size). **f**, Pseudotime analysis of the integrated scRNA-seq data with trajectory (grey line). **g**, Gene cluster analysis of our expression data (coloured by Z-scores) identifies four main branches of co-expressed genes across the different clusters (cl., gene cluster; *n* represents the number of genes in the cluster). Selected GO terms and Kyoto Encyclopedia of Genes and Genomes (KEGG) annotations enriched in each gene cluster are indicated where present. Mosquito schematics in **a** are adapted from ref. 36, CC BY 4.0 and parasite drawings were adapted with permission from ref. 24, Elsevier.

zero—in later stages. The two mosquito-stage rRNAs, S1 and S2 (ref. 34), were expressed throughout oocyst growth, with S1 starting in Took and S2 in day 2 oocyst.

Combined, these gene expression networks provide significant information on the key developmental transitions that parasites undergo in the mosquito midgut, shedding light into the biology of these understudied stages.

## PfATP4 and PfLRS regulate oocyst growth

To assess the functional role of factors emerging from the gene network analysis, we prioritized candidates from different gene clusters that have known potent inhibitors. We selected PfATP4, an Na$^+$-ATPase that is essential for sodium ion efflux (gene cluster 2), which is inhibited by the drug cipargamin (CIP (original name NITD609)), and the

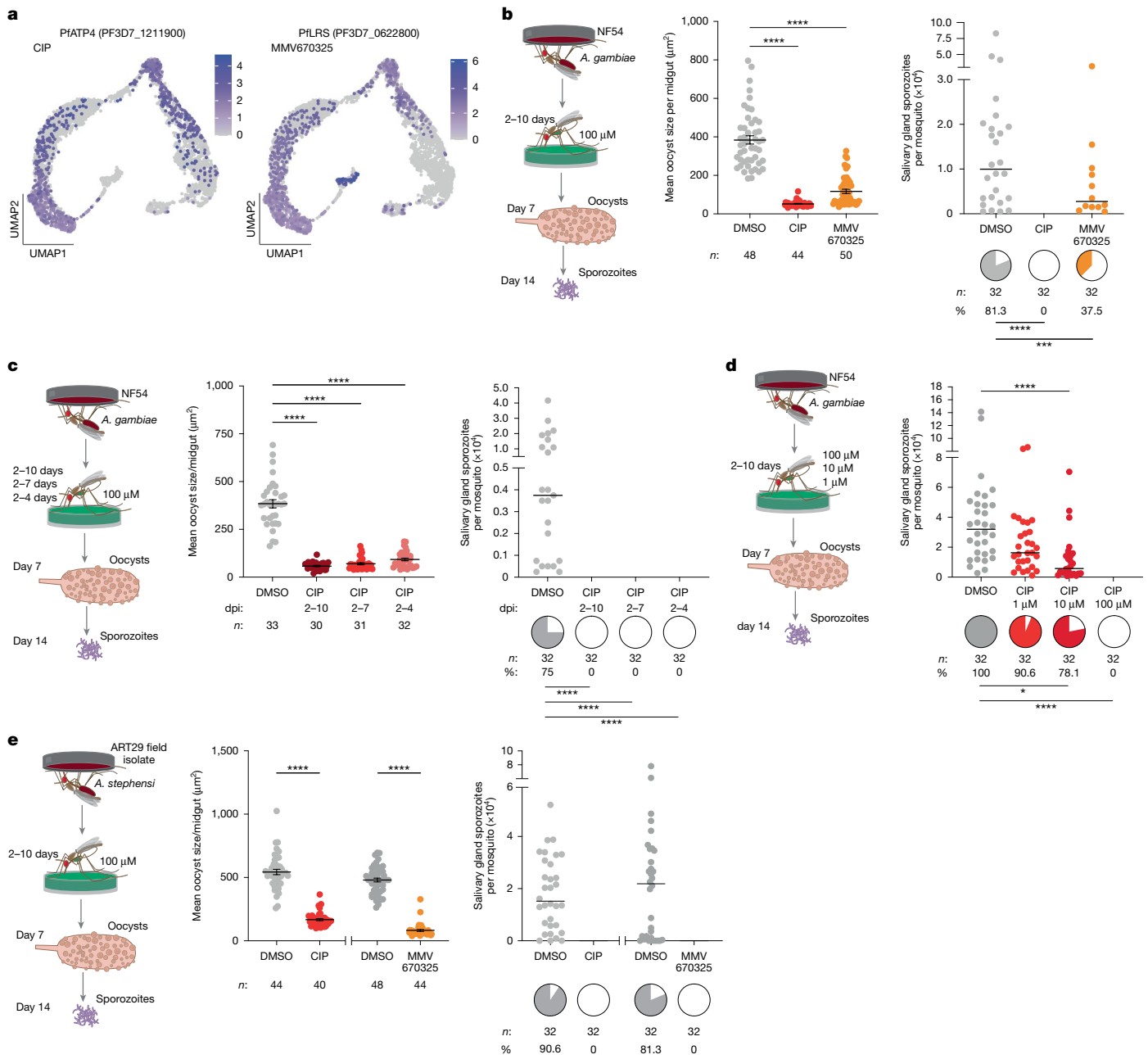

**Fig. 2 | Functional analysis of candidate genes. a**, Expression profile of PfATP4, target of CIP, and PfLRS, target of MMV670325. **b**, Left, *A. gambiae* infected with NF54 parasites were exposed to CIP and MMV670325 via sugar solution at 100 μM delivered from 2 dpi to 10 dpi. Drug ingestion reduced oocyst size at 7 dpi (middle; $P < 0.0001$), subsequently leading to reduced sporozoite prevalence (right; $P < 0.0001$ and $P = 0.008$) in salivary glands at 14 dpi. **c**, Shortening CIP exposure to 2–7 dpi and 2-4 dpi (left) also reduced oocyst size at 7 dpi (middle; $P < 0.0001$), subsequently leading to no sporozoites in the salivary glands by 14 dpi (right; $P < 0.0001$). **d**, Reducing CIP dosage from 100 μM to 10 and 1 μM in *A. gambiae* infected with NF54 parasites led to a dose-dependent reduction in sporozoite prevalence (10 μM versus DMSO: $P = 0.0108$; 100 μM CIP versus DMSO: $P < 0.0001$) and intensity (10 μM CIP versus DMSO: $P < 0.0001$). **e**, Exposure to CIP and MMV670325 in *A. stephensi* mosquitoes infected with ART29 field isolate (left) resulted in comparable reduction in oocyst size at 7 dpi (middle; $P < 0.0001$) and no sporozoites in the salivary glands at 14 dpi (right; $P < 0.0001$). Number of mosquitoes dissected (*n*) and prevalence of infection (%) are indicated for each sample. Oocyst size (cross-sectional area, μm²), averaged per midgut, is represented as mean ± s.e.m., sporozoite intensity is represented as median and both are compared using Kruskal–Wallis test and Dunn's correction (**b**–**d**) or Mann–Whitney two-tailed test (**e**). Infection prevalences represented by pie charts are compared with Fisher's exact test, two-tailed. Data are pooled from two independent infections. *$P < 0.05$, **$P < 0.01$, ***$P < 0.001$, ****$P < 0.0001$. Schematics in **b**–**e** are adapted from ref. 36, CC BY 4.0.

leucine-tRNA ligase (PfLRS, gene cluster 1), which is inhibited by the compound MMV670325 (original name AN6426) (Figs. 1g and 2a). We provided these drugs to infected female mosquitoes via sugar solution, from the onset of oocyst formation (2 dpi) until 10 dpi (Fig. 2b). Although neither drug reduced oocyst intensity or prevalence (Extended Data Fig. 4a), both CIP and MMV670325 considerably impaired oocyst growth. As a result, we detected no sporozoites in the salivary glands of CIP-treated mosquitoes at 14 dpi and observed a 54% reduction in sporozoite prevalence after MMV670325 ingestion relative to controls (Fig. 2b). Shorter exposure times of CIP (2–4 dpi and 2–7 dpi) gave similar results, suggesting that regulation of sodium efflux is essential during the early phases of oocyst growth,

and the effects were dose-dependent (Fig. 2c,d and Extended Data Fig. 4b–d).

Similar results were obtained with both drugs in a different mosquito–parasite combination (*Anopheles stephensi* infected with *P. falciparum* ART29, an artemisinin-resistant field isolate from Cambodia), and also when exposing female *A. gambiae* to CIP after infection with *P. falciparum* P5, a polyclonal field isolate from Burkina Faso[25] (Fig. 2e and Extended Data Fig. 4e,f). These results validate PfATP4 and PfLRS as essential for oocyst development in *P. falciparum*.

Two additional targets—elongation factor 2 PfeEF2 (cluster 12) and the lactate/H⁺ transporter PfFNT (cluster 7)—showed smaller effects when mosquitoes were exposed to sugar solutions of their respective inhibitors, MMV643121 and MMV007839 (Fig. 1g and Extended Data Fig. 4g,h). MMV643121 slightly decreased oocyst size at 7 dpi but did not affect sporozoite intensity or prevalence, whereas MMV007839 had no effect on oocyst size but significantly reduced sporozoite intensity, indicating a possible role for this transporter in sporozoite segmentation or salivary gland invasion (Extended Data Fig. 4g,h).

These results highlight how our scRNA-seq data can be systematically mined to uncover essential targets for parasite development. The genes validated here could be targeted in novel mosquito-based malaria control strategies, as previously demonstrated using cytochrome b inhibitors[35,36].

## PfSIP2 ensures hepatocyte invasion

In the final branch, along with rhoptry and cytoskeletal genes, we also detected the ApiAP2 transcription factor PfSIP2 (gene cluster 10) (Figs. 1g and 3a). In asexual blood stages, PfSIP2 has an essential role probably in daughter merozoite formation[37], yet its role during mosquito stages remains unknown. To test its function, we generated a conditional PfSIP2-knockdown (PfSIP2-cKD) line integrating the TetR-DOZI aptamer system at the C terminus (Extended Data Fig. 5a,b). Removal of anhydrotetracycline (ATC) in asexual blood stage cultures led to parasite death, confirming the essentiality of PfSIP2 (ref. 37) (Extended Data Fig. 5c). We adapted this system to mosquito stages by providing ATC throughout gametocyte cultures and in the mosquito daily sugar solutions, and induced gene knockdown by withdrawing the compound immediately before mosquito infection (Fig. 3b). PfSIP2-cKD parasites were less infectious to mosquitoes than wild-type (WT) co-cultured ones, but this is often observed in mosquito infections with transgenic parasites that have undergone bottlenecks during the selection process. When we compared PfSIP2-cKD +ATC and −ATC groups, we found no differences in terms of oocyst intensity, prevalence and size at 7 dpi, nor in sporozoite intensity and prevalence at 14 dpi (Extended Data Fig. 5d–f). ATC withdrawal, however, severely impaired the ability of PfSIP2-cKD sporozoites to invade primary human hepatocytes, inducing a remarkable decrease in both invasion efficiency (67% reduction) and the formation of exoerythrocytic forms (96% reduction) (Fig. 3c,d).

PfSIP2 expression is not detected in salivary gland sporozoites[5,21,38] (although we cannot rule out expression below detection limits), suggesting that hepatocyte infection is a function of its expression in oocysts. Regardless, our data reveal that the downstream targets of PfSIP2 could provide new candidates for transmission-blocking strategies.

## Parasites interact with progenitor cells

Sequencing of both parasites and mosquito cells gave us the opportunity to identify possible interactions between the two organisms. We first analysed the mosquito datasets from all four time points and the two treatments (ds*GFP* and ds*EcR*). After quality control using read and gene counts, mitochondrial read percentages, and complexity score (Extended Data Fig. 6a and Supplementary Table 1), we successfully profiled 55,789 high-quality midgut cells. Cells from each sample were

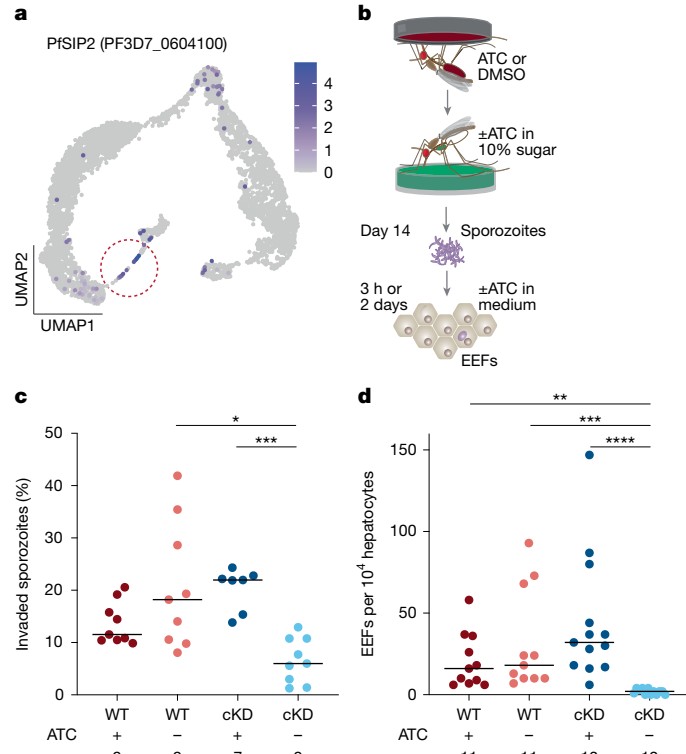

**Fig. 3 | PfSIP2 knockdown impairs hepatocyte infection. a**, Expression of PfSIP2 (PF3D7_0604100) is highest in segmenting sporozoites within late oocysts (circled in the UMAP). **b**, Schematic representation of primary human hepatocyte infections with WT or PfSIP2-cKD parasites. **c,d**, Knockdown by ATC (500 nM) withdrawal impaired sporozoite invasion of primary hepatocytes at 3 hpi (**c**; PfSIP2-cKD, no ATC versus WT, no ATC: *P* = 0.0147; PfSIP2-cKD, +ATC versus no ATC: *P* = 0.0009) and reduced formation of exoerythrocytic forms (EEFs) at 2 dpi (**d**; PfSIP2-cKD, no ATC versus WT, +ATC: *P* = 0.0048; PfSIP2-cKD, no ATC versus WT, no ATC: *P* = 0.0004; PfSIP2-cKD, +ATC versus no ATC: *P* < 0.0001). Kruskal–Wallis test, Dunn's correction. Data are represented as median. *n* indicates the number of independent wells across three biological replicates. Schematics in **b** are adapted from ref. 36, CC BY 4.0.

then integrated using reciprocal PCA (RPCA) to correct for treatment and batch effects, followed by dimensionality reduction and clustering based on transcriptional profiles. We annotated the 16 resulting clusters on the basis of marker gene expression and gene ontology (GO) terms (Fig. 4a, Extended Data Fig. 6b,c and Supplementary Table 5; a detailed description is provided in the Supplementary Note). The major clusters included progenitor cells (intestinal stem cells/enteroblast (ISC/EB, headcase-positive)); posterior enterocytes (pEC, nubbin-positive); anterior enterocytes (aEC, sugar transporter-positive); enteroendocrine cells (EE, prospero-positive); visceral muscles (VM, myosin-positive); and proventriculus cells (PV, eupolytin-positive (also called cardia)). A group of prospero-positive cells with mixed markers were annotated as EE-like. Of note, comparing ds*GFP* samples with ds*EcR* samples revealed limited effect on mosquito transcriptomes, with fewer than ten genes being differentially expressed at each time point (Extended Data Fig. 6d and Supplementary Table 6). By contrast, time post blood meal strongly affected gene expression across all major cell types (Extended Data Fig. 6e and Supplementary Table 7).

To identify mosquito–parasite interactions, we focused on the 36 hpi time point when parasites are crossing the epithelial layer. As ds*GFP* and ds*EcR* samples presented very similar mosquito and parasite transcriptomes, we pooled cells from these two groups (Extended Data Fig. 2a and 6d and Supplementary Table 6). We reasoned that a parasite captured inside or in strong association with a midgut cell

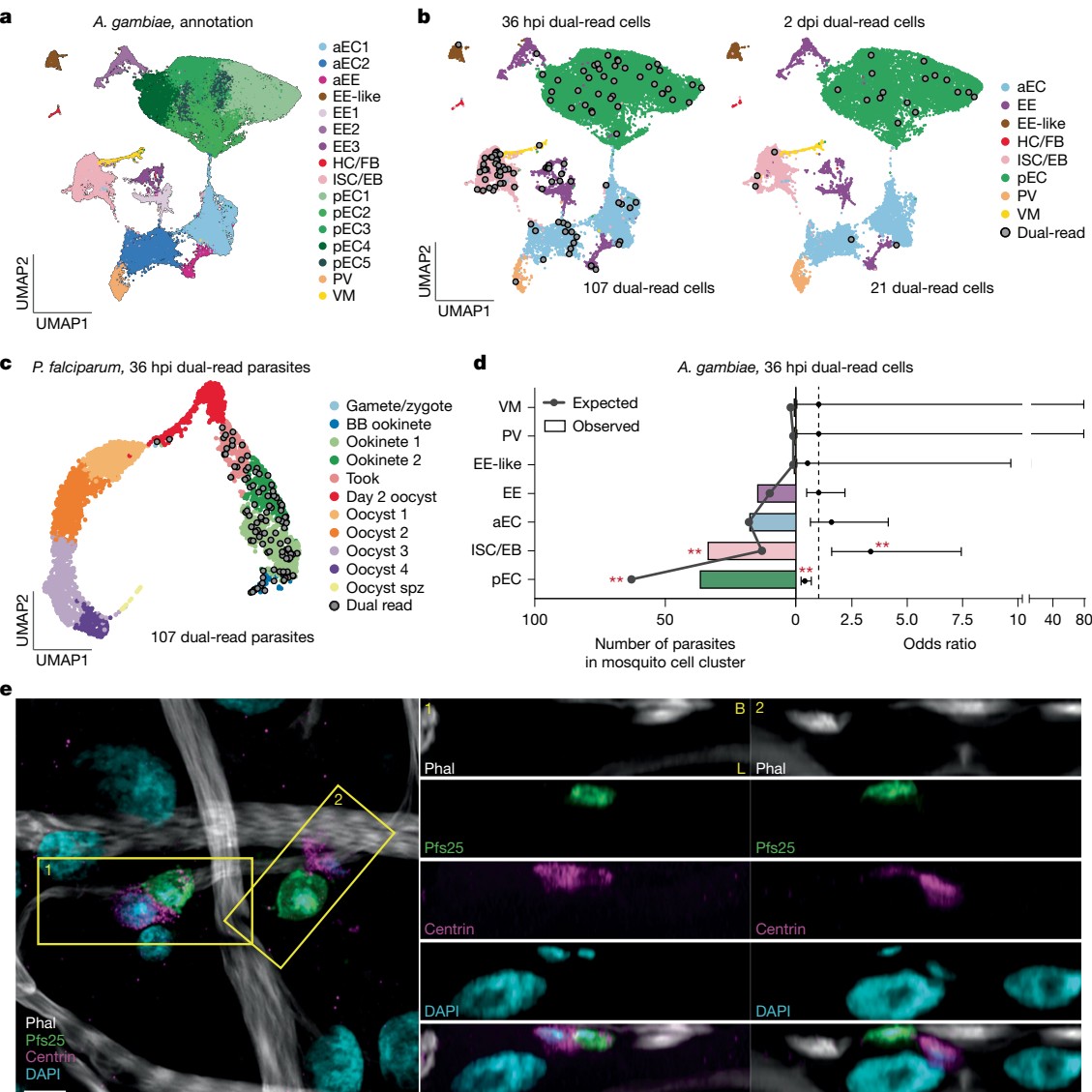

**Fig. 4 | Parasite interactions with progenitor cells during midgut traversal.** **a**, UMAP of *A. gambiae* midgut cells across all 4 time points forms 16 cell clusters, representing different cell types: aECs, EE, haemocyte and fat body cell (HC/FB), ISC/EB, pEC, PV, VM and a cluster with mixed markers that we annotate as EE-like. **b**, UMAP plots of *A. gambiae* coloured by cell type and overlaid with dual-read cells (cells with the same mosquito and parasite barcodes, indicated by grey circles) at 36 hpi (left) and 2 dpi (right). **c**, UMAP plot of *P. falciparum* from this study overlaid with dual-read parasites (grey circles) at 36 hpi. **d**, Distribution of 107 dual-read parasites over 8 independent samples in mosquito cell types at 36 hpi with the observed versus expected numbers under the random distribution (left *x* axis) and the resulting odds ratio ± 95% confidence interval (right *x* axis). Two-sided Fisher's exact test with Benjamini–Hochberg multiple comparison correction. **\*\*FDR = 0.003. **e**, Confocal imaging shows that some parasites are inside of, or interact with, progenitor cells at 36 hpi. Left, maximum intensity projection of a 2.31 µm *z*-stack (0.21 µm between each plane) showing interactions between parasites (Pfs25, green) and progenitor cells (centrin, magenta). Two orthogonal slices (yellow boxes) show parasites possibly inside (1) and adjacent (2) to progenitor cells on the basal side (B, top) of the midgut. L marks the luminal brush border. Similar interactions were observed across three independent infections. Nuclei were stained with DAPI (cyan) and muscle actin was stained with phalloidin (Phal, white). LSM Plus processing was used. Scale bars, 5 µm.

would have the same cellular barcode as the mosquito cell. Despite the fleeting nature of crossing[11–13], we detected 107 parasites that shared a barcode with a mosquito cell (defined here as dual-read cells) out of the 1,017 sequenced at this time point (10.5%) (Fig. 4b and Extended Data Fig. 7a). This percentage is significantly higher than the 3.9% observed at 2 dpi (21 dual-read parasites out of 533 total, Fisher's exact test, *P* value < 0.001), when parasites have mostly transformed into extracellular oocysts.

Of note, dual-read parasites appeared to interact preferentially with the ISC/EB midgut cluster (Fig. 4b–d). Indeed, 32.7% of dual-read cells (3.3% of total parasites) included a progenitor cell, although this cluster represents only 12.1% of all mosquito cells at this time point

(Fig. 4d, odds ratio 3.3, false discovery rate (FDR) = 0.003). This enrichment was not due to lower mRNA content in ISC/EB cells, which might artificially increase parasite mRNA detection rates, and was not observed at 2 dpi (Extended Data Fig. 7b,c). Parasites were also less likely to be associated with pECs than expected by chance (Fig. 4d, odds ratio 0.37, FDR = 0.003). In differential expression analyses, no protein coding genes of midgut progenitor cells were affected by the presence of an interacting parasite (FDR = 1), and similarly, parasites interacting with a progenitor cell showed no difference in their transcriptomes relative to those of non-interacting parasites (FDR = 1), although these results may be due to the limited number of dual-read cells.

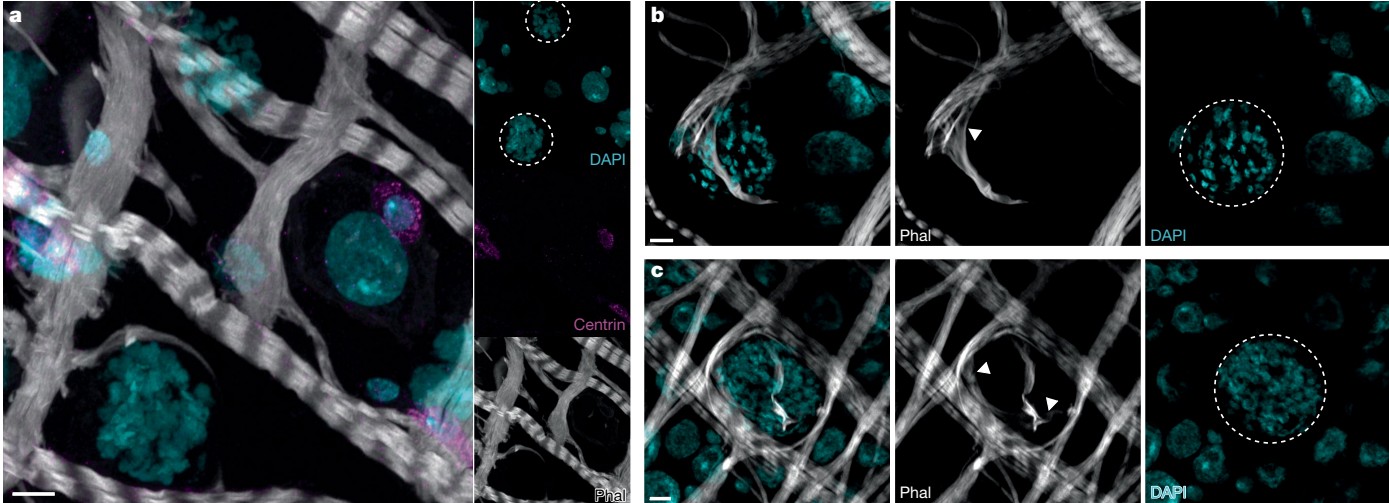

**Fig. 5 | Late oocysts interact with visceral muscles. a**, Maximum intensity projection of confocal images showing the absence of progenitor cell accumulation (centrin, magenta) around 7 dpi oocysts (dashed circle; Supplementary Video 2). **b,c**, Two representative images showing muscle fibres (phalloidin, grey) forcefully stretched (arrowhead) around a large oocyst (dashed circle) (**b**) or encasing an oocyst (**c**) (Supplementary Video 3). Infections were performed in triplicate. Projections of 6.72 μm (**a**) and 2.94 μm z-stacks (**b,c**) are presented. Scale bars, 5 μm.

To confirm these findings, we performed confocal microscopy using a centrin antibody that labels progenitor cells, as determined by their basal location, smaller triangular shape and colocalization with the known progenitor marker armadillo[39] (Extended Data Fig. 7d). We observed close interactions between parasites and progenitor cells, with 10.1% of parasites (35 out of 347) appearing to be inside or in close contact with an ISC/EB (Fig. 4e, regions 1 and 2), or oriented towards one (Extended Data Fig. 7e,f). This number exceeds the 3.3% observed by scRNA-seq, suggesting that the single-cell isolation process preferentially preserved the strongest interactions. Consistent with previous studies[12,15,40], we also observed parasites near pECs that were either extruded or positive for the parasite surface protein Pfs25 (Extended Data Fig. 7e,f and Supplementary Video 1), potentially accounting for the reduced number of dual-read pECs associated with parasites.

We confirmed these findings by confocal microscopy in different mosquito–parasite combinations, observing close interactions between parasites and progenitor cells in 10.9% (56 out of 512) of parasites of the field isolate *P. falciparum* P5 in *A. gambiae* and in 8.71% (31 out of 356) of a different NF54 clone (referred to here as NF54_H) in a recently colonized *Anopheles coluzzii* strain[41] (Extended Data Fig. 7g,h).

Combined, these data reveal a preferential interaction, conserved across *Anopheles* species and *P. falciparum* isolates, between parasites and mosquito progenitor cells during traversal of the midgut epithelium, as ookinetes transform into oocysts.

## Late oocysts are associated with muscles

Notably, midgut progenitor cells were not observed to interact with ookinetes in the rodent-infecting *Plasmodium berghei* model, but were instead associated with late oocysts, where their expansion eventually led to parasite death[42]. However, we did not find evidence of accumulation of centrin-positive cells near oocysts at 7 dpi (Fig. 5a), and conversely detected a decrease in the proportion of progenitors over time (Supplementary Table 5).

In our confocal microscopy analyses, however, we identified phalloidin-stained visceral muscles closely wrapped around late oocysts, with muscle fibres extending beyond their usual width (Fig. 5b,c and Supplementary Videos 2 and 3). This interaction was conserved in several other mosquito–parasite combinations: *A. gambiae*–*P. falciparum* P5; *A. coluzzii*–NF54_H; and *A. stephensi*–ART29 (Extended Data Fig. 8a–c). Physical interactions with muscle cells may potentially tether oocysts to the midgut while minimizing disruption of the epithelial layer, as discussed below.

## Discussion

By focusing on developmental stages that had previously escaped large-scale analysis, our study provides a comprehensive reconstruction of the events that characterize the *P. falciparum* transmission cycle. This rich volume of information expands the repertoire of potential targets for preventing parasite transmission in strategies based on transmission-blocking drugs and vaccines[43] or mosquito-targeted antimalarials[35,36]. Indeed, our functional analyses validate parasite factors (PfATP4, PfLRS and PfSIP2) as essential for parasite growth and the acquisition of infectivity to human hepatocytes. We also reveal pathways that are potentially involved in the ookinete–oocyst transition, a possible role of PfEMP1 in adherence of newly formed oocysts to the midgut, and processes that fuel oocyst development, although these findings will require validation in future studies. Of note, our results demonstrate that much of the core machinery necessary for daughter cell formation (inner membrane complex, basal complex and other components) is conserved between blood stage parasites and oocysts, despite the marked difference in scale of this process between these stages (tens of merozoites versus thousands of sporozoites).

The parallel sequencing of both parasites and mosquito cells enabled the identification of a previously unappreciated interaction between parasites and midgut progenitor cells. Ookinete traversal of the midgut epithelium is an asynchronous process that occurs between 24 hpi and 36 hpi. The best supported model for midgut crossing to date suggests that ookinetes initially invade an epithelial cell, then move laterally to adjacent cells before reaching the basal side, often leading to the extrusion of the invaded cells[14,44]. The observation that dual-read parasites contain not only ookinetes but also transforming tooks suggests that this interaction with progenitor cells occurs as parasites reach the basal side, where they transition into oocysts. Although our findings only begin to uncover the mechanisms that govern ookinete exit and transformation into an oocyst, it is plausible that *P. falciparum* uses progenitor cells as landmarks for the basal side of the midgut. Here, parasite exit may be facilitated by the remodelling of tight junctions during division of ISCs[45,46], although several alternative hypotheses

are possible. It is likely that these interactions occur after ookinetes have rapidly escaped extruding pECs, as observed both in this study and in several others[14,15,40], and in alignment with our observation of a slightly reduced number of parasite-associated pECs.

In our analysis, *P. falciparum* oocysts were found to interact tightly with muscle cells. This interaction may be the result of the specific co-evolutionary trajectories of human malaria parasites with human-biting mosquitoes. Indeed, *P. falciparum* possesses mechanisms to minimize the damage inflicted to its *Anopheles* vectors[24,47], including a low number of ookinetes crossing the midgut epithelium. By contrast, rodent malaria parasites have high parasite loads that induce substantial epithelial damage and overall fitness costs[48,49], and their oocysts are deeply embedded in the epithelium[50], probably causing damage and consequent proliferation of ISCs[42]. A close association with muscle cells might enable *P. falciparum* oocysts, which on the contrary jut farther out of the midgut[50], to limit epithelial damage by tethering growing parasites in a less invasive way. Further research will clarify the biological relevance of these mosquito–parasite interactions.

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

## Methods

### Rearing of *Anopheles* mosquitoes

*A. gambiae* (G3 strain), *A. stephensi* (SDA-500), and recently colonized *A. coluzzi*[41] mosquitoes were reared in an insectary maintained at 27 °C, 70–80% relative humidity, on a 12 h:12 h light:dark cycle. Larvae were fed on fish food and adults were provided with 10% w/v glucose solution ad libitum. Adult mosquitoes were fed with purchased human blood (Research Blood Components) to induce egg laying.

### dsRNA production and microinjection

PCR fragments of *eGFP* control (495 bp) and *EcR* (AGAP029539; 435 bp) were amplified from the plasmids pCR2.1-eGFP and pCR2.1-EcR as previously described[51]. PCR product was used to transcribe dsRNA using the Megascript T7 transcription kit (Thermo Fisher Scientific), per the manufacturer's protocol. The resulting dsRNA was purified via phenol-chloroform and diluted to a concentration of 10 µg µl$^{-1}$. Microinjections were performed as detailed previously[24], with adult mosquitoes, randomly assigned to each group, being injected one day post-eclosion and given an infectious blood meal 3 days post-injection.

### EcR knockdown confirmation

Resulting knockdown confirmation post-injection was performed as previously described[24]. Pools of 6–8 infected female mosquitoes were aspirated at 24 hpi and transferred to TRI reagent (Thermo Fisher Scientific). RNA was extracted, DNase treated, quantified and reverse transcribed. cDNA from four biological replicates was diluted tenfold and run in duplicate for quantitative PCR with reverse transcription. Gene-specific primers were used, and relative quantification was performed with the ribosomal gene *Rpl19* as the reference[52] (Supplementary Table 8). RNA extraction from one replicate of the single-cell experiments failed and were thus repeated with samples collected at 2 dpi. For phenotypic EcR knockdown confirmation, individual mosquito ovaries were also collected 7 dpi in 80% ethanol for later manual egg counts.

### Culturing of *P. falciparum* parasites

*P. falciparum* strains used in this study include: (1) NF54 originally provided by C. Barillas-Mury from BEI Resources (MRA-1000); (2) NF54_H, highly infectious to mosquitoes, from the laboratory of M. Delves; (3) an artemisinin-resistant, Cambodian field isolate ART29 (ref. 53); and (4) a Burkina Faso polyclonal isolate P5 (refs. 25,35). All strains were identified as *P. falciparum* parasites by nested PCR or whole genome sequencing, and confirmed mycoplasma free. Asexual stages of parasites were cultured at 37 °C between 0.2 and 2% parasitaemia in human erythrocytes at 5% haematocrit (Interstate Blood Bank) using RPMI medium 1640 supplemented with 25 mM HEPES, 10 mg l$^{-1}$ hypoxanthine, 0.2% sodium bicarbonate, and 10% heat-inactivated human serum (Interstate Blood Bank) under a gas mixture of 5% $O_2$, 5% $CO_2$, balanced $N_2$. Stage V female and male gametocytes were induced by raising parasitaemia over 4% and incubating cultures for 14–20 days with daily medium change.

### *P. falciparum* infections of *Anopheles* mosquitoes

Female mosquitoes were blood-fed on a mix of *P. falciparum* gametocyte culture, human red blood cells and human serum (ratio 1:3:6) via heated membrane feeders 3–5 days post-eclosion, and introduced into a custom-built glove box (Inert Technology) as previously described[24]. Unfed mosquitoes were removed, and remaining mosquitoes were kept on 10% glucose solution ad libitum, with added ATC or compound (see below). At dissection time points, mosquitoes were aspirated into 80% ethanol, frozen for 10 min and transferred to phosphate-buffered saline (PBS) on ice.

### Generation of single-cell suspension

Eighty midguts from *P. falciparum*-infected mosquitoes injected with ds*GFP* or ds*EcR* were dissected at 36 hpi, 2 dpi, 4 dpi and 7 dpi across 4 biological replicates in ice-cold PBS. Any remaining blood meal was removed before storing the dissected midguts in ice-cold PBS. Dissections were completed within 1 h to preserve the viability of parasite and mosquito cells. Midguts were pooled into 200 µl of PBS with 1 mg ml$^{-1}$ elastase (Sigma-Aldrich E7885) and 1 mg ml$^{-1}$ collagenase IV (Thermo Fisher Scientific 17104019), and incubated at 30 °C, 300 rpm for 30 min in a shaking heat block. To facilitate the isolation of single cells, samples were disrupted every 10 min by pipetting with low protein-binding tips.

Samples were assessed using trypan blue staining to quantify live, single midgut cells during protocol optimization. This included systematically testing variables such as digestion time, temperature, tissue preparation (intact or minced gut) and types of pipette tip. On average, approximately 400 live, single cells were consistently recovered per mosquito midgut, while many cells remained in clumps. Given the low number of parasites (~20–30) relative to mosquito cells (~5,000) in a midgut, rather than further optimizing for complete dissociation of the midgut, we chose to enrich the parasite-to-mosquito cell ratio by removing cell clumps as they were composed mostly of midgut cells. Therefore, the cell suspension was sequentially filtered through 70 µm, 40 µm and 20 µm cell strainers (pluriSelect), with each strainer being immediately washed with 200 µl of 1× PBS to collect any adherent cells. The 20 µm cell strainers were not used for the 7 dpi samples, as oocysts at this time point are, on average, over 25 µm in diameter[24]. Based on the final dataset, this approach yielded an approximate 1:16 parasite-to-mosquito cell ratio. Given an estimated 5,000 midgut cells and 30 oocysts per gut—an expected ratio of about 166:1—our protocol represents a tenfold enrichment of parasites.

We chose not to collect samples beyond 7 dpi, as oocysts larger than 30 µm may clog the 10x Chromium chip. We chose not to use fluorescence-activated or magnetic-activated cell sorting as they both lengthen the isolation protocol and cause stress, reducing cell viability. The final single-cell suspension was pelleted, resuspended in 20 µl of PBS, and cell concentration and viability were determined by haemocytometer count with trypan blue staining.

### Single-cell library preparation and sequencing

Single-cell libraries were prepared following the Chromium Next GEM Single Cell 3′ Reagents Kit v.3.1 (Dual Index) User Guide (RevC). In brief, single-cell suspensions were diluted to target a recovery of 4,000 cells per sample, then mixed with reverse transcription reagents and barcoded gel beads. The mixtures were loaded into wells containing partitioning oil on a 10x Chromium chip to generate single-cell emulsions. Samples from ds*GFP* and ds*EcR* mosquitoes collected at the same time point were loaded into two separate wells on the same chip. In total, 16 chips were used across 4 time points and 4 biological replicates. Within each emulsion droplet, individual cells were lysed, mRNA molecules were captured, and cDNAs were generated with cell-specific barcodes. Barcoded cDNAs were pooled, then amplified, fragmented and purified using SPRIselect, followed by ligation with Illumina adaptors and sample-specific barcodes. Each sample was assessed using an Agilent Bioanalyzer to ensure library integrity and quantified by qPCR using i7 and i5 Illumina primers, following the protocol from Universal Kapa Library Quantification Kit (Roche). The cDNA libraries from each sample were then mixed in equal proportions, spiked with 1% phage cDNA (PhiX, Illumina). The proportion of parasite reads relative to mosquito reads was estimated using an initial Illumina iSeq 100 run, which showed that around 0.05%, 0.1%, 0.6% and 1.3% reads mapped to parasites at 36 hpi, 2 dpi, 4 dpi and 7 dpi, respectively. To obtain sufficient coverage of parasite transcripts, final sequencing was performed across all lanes of two NovaSeq S4 flow cells (Broad Institute), with an expectation of

1,974–51,316 reads per parasite ranging from 36 h to 7 dpi (see Supplementary Note for detailed calculation).

## scRNA-seq data processing and analysis of *P. falciparum*

FASTQ files from each sample originating from different flow cells were concatenated and mapped to the genomes of *P. falciparum* 3D7 (PlasmoDB.org, v.58)[54] using 10x Cell Ranger software v.7.0.1 (ref. 55). The resulting count matrix for each sample was processed and filtered using Scanpy (v.1.9.1) in Python (v.3.10)[56]. Based on the individual sample profile, low-quality cells, dead cells and empty droplets were removed according to the number of reads (unique molecular identifier (UMI) count), number of genes (gene count) and proportion of mitochondrial reads per cell (Supplementary Table 1). The count matrix from each sample was merged into a single AnnData object by Scanpy, and doublets were removed running Scrublet software in Python before conversion to Seurat using SeuratDisk's function Convert[57]. After quality control, the average number of reads per cell ranged from 649 at 36 hpi to 9,043 at 7 dpi. Pseudobulk differential expression analyses were initially conducted treating the samples as independent observations for comparison. The pseudobulk count matrix was generated for each sample by summing raw gene counts across all cells in Python with Pegasus. Several comparisons were conducted—between each time point in parasites from ds*GFP*-injected mosquitoes as well as between parasites from ds*EcR*- and ds*GFP*-injected mosquitoes of the same time point—using a Python wrapper for the DESeq2 package in R[58]. Significance was set as $log_2$ fold change greater than 1 or less than −1, and an adjusted *P* value below 0.05. Functional enrichment analysis of upregulated or downregulated gene lists was performed using GoProfiler[59].

A standard single-cell analysis pipeline was then performed with Seurat and dependent packages in R 4.3.2 in RStudio 2023.9.1.494 (ref. 60). Highly variable genes were identified with the FindVariableFeatures function before data were scaled (ScaleData, default settings) and a PCA plot was generated based on these features only. Next, high-quality parasite cells were integrated with existing datasets, including blood bolus ookinetes at 24 hpi and 48 hpi, gametes, zygotes and ookinetes from 2–20 hpi, and sporozoites released from oocysts[21,22] at 12 dpi. In brief, Seurat objects (v.5) were created for external datasets and reads were normalized to 10,000 transcripts, before the three Seurat objects were merged. Variable features were identified with default parameters of the FindVariableFeatures function, the data were scaled (ScaleData), and PCA was performed on the first 50 dimensions (RunPCA, default settings). Integration to correct for batch effect and differences in single-cell methodology was conducted using RPCA, which is the preferred approach when minimal overlap between datasets is expected. Subsequently, a UMAP was generated including the top 12 principal components through the RunUMAP function, with parameters set to min.dist = 0.4, and repulsion.strength = 2. Cell clusters were defined by the Louvain algorithm at a resolution of 0.5. Cluster markers were identified using the FindAllMarkers function, focusing only on positive markers with a $log_2$ fold change above 1. Genes with FDR values below 0.01 were utilized to label each cluster.

The Seurat object was converted into a cds object in Monocle3 (ref. 61), and the trajectory analysis was performed using the learn_graph function (close_loop = F, ncenter=500). Pseudotime analysis was conducted by setting the node in the gamete–zygote cluster as the start point. Gene network analysis focused on our data and was performed using the graph_test function (neighbor_graph = "principal_graph"), to identify co-expressed genes that change as a function of pseudotime (FDR < 0.05). Co-expressed genes were further clustered by the find_gene_modules function with a resolution of 0.0023. The expression of all genes within each gene cluster was aggregated using the aggregate_gene_expression function. Enriched GO terms were determined by analysing all the genes within a gene cluster in GoProfiler[59]. Differential expression of protein coding genes was compared between dual-read parasites at 36 hpi and parasites that do not share a barcode with a mosquito cell, binning for parasite cluster, using logistic regression. A subsequent analysis focused on dual-read parasites only at 36 hpi, and compared those interacting with an ISC/EB to the remainder.

## Mosquito scRNA-seq data processing and analysis

All FASTQ files from the different sequencing runs were concatenated and mapped to the *A. gambiae* PEST genome (VectorBase.org, v.58)[54], using the 10x Cell Ranger software v.7.0.1 (ref. 55). Quality control was initially performed in Python using Scanpy based on the number of reads (UMI count), number of genes (gene count), percentage of reads mapping to the mitochondrial genome and cell complexity (Supplementary Table 1). Cells with a complexity score (ratio of number of transcripts by genes logged) below 0.7 or more than 30% reads mapping to the mitochondrial genome were removed[62–67]. The mitochondrial threshold was chosen on the basis of published scRNA-seq studies of *Anopheles* haemocytes[62] and *Aedes* midguts[63], noting that a similar threshold (25%) was used by two scRNA-seq studies on the midgut of *Drosophila melanogaster*[64] and *Culex tarsalis*[65], whereas snRNA studies used lower cutoffs due to inherent differences in the technology[66,67]. The resulting count matrix from each sample was merged into a single AnnData object by Scanpy, before conversion to Seurat using SeuratDisk's function Convert. Doublet removal was performed in R using scDblFinder[68]. Quality metrics indicate that mitochondrial percentage is lower in 36 h and 2 dpi samples compared to the 4 dpi and 7 dpi ones, probably owing to a combination of biological factors, including time after blood feeding, age and infectious stages (ookinetes or young oocysts versus large oocysts).

After quality control, mosquito cells from each time point were processed using a similar workflow as for parasite analysis. The top 25% variable genes were identified, the data were scaled, and PCA was performed computing the first 100 PCs. Integration was conducted using RPCA with the highest k.weight (64) to correct batch effect while avoiding overcorrecting for biological variation between time points and treatments. The UMAP was calculated using the first 56 PCs, and cluster identification was based on the Louvain algorithm with a resolution of 0.3. Positive markers were identified using the FindAllMarkers function with a $log_2$ fold change greater than 1. Proventriculus, anterior and posterior midgut scores were calculated based on *A. gambiae* midgut bulk RNA-seq results[69]. Clusters were then annotated by identifying the functional enrichment of each cluster's marker genes as well as cross-referencing cluster markers from four references: the Fly Cell Atlas, the *D. melanogaster* gut single-cell study, the *Aedes aegypti* gut single-cell study, and the haemocyte single-cell study from *A. gambiae*[62,64,66,67]. *A. gambiae* orthologues were identified using VectorBase[54], and all significant cluster markers with an orthologue specific to a cell type were used for cluster annotation.

Pseudobulk analyses were performed to assess the effect of either *EcR* knockdown or time post-infectious blood meal on the midgut transcriptome using AggregateExpression from the Seurat package. To assess the effect of *EcR* knockdown on transcription, differential expression analyses were performed using DESeq2 (ref. 58) by pooling cells from each cell type at each time point. To evaluate the effect of time on transcription, only cells from ds*GFP* mosquitoes were used to compare between consecutive time points. Functional enrichment analyses were performed using GoProfiler[59] when more than 20 genes were significant. Dual-read differential expression analyses of protein coding genes were performed using logistic regression framework comparing dual-read cells at 36 hpi with cells not interacting with a parasite at the same time point, binning for dsRNA treatment and annotation. The analysis was repeated by comparing protein coding gene expression between dual-read ISC/EB cells at 36 hpi and ISC/EBs not interacting with a parasite.

## Post-infection exposure assays via sugar feeding

Candidate target selection for drug exposure was based on two criteria: (1) belonging to a distinct gene cluster in our gene model; and (2) having

known potent activity against the candidate target in the *P. falciparum* blood stage. Four drugs were selected: cipargamin (ChemPartner), MMV670325, MMV643121 and MMV007839 (refs. 70–73) (Medicine for Malaria Venture). Stock solutions were made in DMSO at 100 mM, stored at −20 °C and diluted 1:1,000 in 10% glucose solution shortly before sugar change. As the bioavailability of these compounds in mosquitoes is unknown, a single high concentration of 100 μM was chosen initially as it is close to the solubility limit while keeping DMSO content to 0.1%. Mosquito compound exposure through sugar feeding was performed similarly to previously described[53]. In brief, sugar solutions containing compounds or corresponding controls (0.1% DMSO) were randomly assigned to a mosquito group and replaced every day from 2 dpi to 10 dpi (unless specified). For oocyst counts, midguts were collected at 7 dpi and stained in 0.2% (w/v) mercurochrome (Sigma-Aldrich). Midguts were imaged using an Olympus Inverted CKX41 microscope, and oocyst number and mean size were calculated using OocystMeter[74]. For salivary gland sporozoite quantification, salivary glands of individual female mosquitoes were dissected at 14 dpi in PBS and counted on a haemocytometer.

### Plasmid construction

The PfSIP2 homology-directed repair (HDR) plasmid was assembled by Golden Gate cloning (New England Biolabs) using purified PCR products of (1) a C-terminal homology region (Cterm-HR) that was codon-altered to avoid repeated CRISPR cuts; (2) a cassette containing haemagglutinin (HA) tags, tet-aptamers[75], a TetR-DOZI fusion protein[75,76], and a human dihydrofolate reductase (hDHFR) selection marker; (3) a 3′ homology region; and (4) a bacterial backbone. Primers used to amplify fragments are listed in Supplementary Table 8 and assembly was confirmed by whole plasmid sequencing (Plasmidsaurus). Two CRISPR–Cas9 guide plasmids were constructed to cut the C terminus of the endogenous PfSIP2. Guide oligonucleotides were annealed and ligated into BpiI-digested pRR216 (ref. 77), which contains SpCas9, a U6 guide cassette, and a yeast dihydroorotate dehydrogenase (DHODH) selection marker. Sequences were confirmed by Sanger sequencing (Psomagen).

### Generation of PfSIP2-cKD parasite line

The HDR and guide plasmids of PfSIP2 were extracted using a Qiagen Maxi kit, and 100 μg of the HDR plasmid was linearized with XhoI, NotI, and ApaL1 at 37 °C overnight. The linearized HDR plasmid and a mixture of the 2 guide plasmids (60 μg each) were sterilized by ethanol precipitation and resuspend in TE buffer. The plasmids were then mixed with 270 μl of Cytomix (120 mM KCl, 0.15 mM CaCl$_2$, 2 mM EGTA, 5 mM MgCl$_2$, 10 mM K$_2$HPO$_4$, 10 mM KH$_2$PO$_4$, 25 mM HEPES, pH 7.6) and transfection by red blood cell loading was performed as described[78]. The parasite line was maintained at 5% haematocrit with 500 nM ATC (Sigma) from the onset of transfection. Drug pressure for the guide plasmid was applied from 6 h to 5 days post-transfection using 1.5 μM DSM1 (Sigma), whereas drug pressure for the HDR plasmid was maintained with 2.5 nM WR99210 hydrochloride (Sigma) starting 6 h post-transfection until integration was confirmed. To do so, genomic DNA was extracted from WT and PfSIP2-cKD lines using the Qiagen Blood & Tissue kit. Each integration site of the double crossover was amplified as well as the whole locus and PCR products were analysed by gel electrophoresis.

### Asexual stage growth assay

ATC was washed out from PfSIP2-cKD parasites three times with RPMI 1640. The resulting parasites were diluted to 0.5% starting parasitaemia at 2.5% haematocrit and treated with either 500 nM ATC, 1.5 nM ATC or 0.025% of DMSO. Parasitaemia was calculated daily by counting Giemsa-stained smears of each technical replicate. The experiment was repeated twice, with four technical replicates for each biological replicate. Data were analysed by two-way ANOVA with Dunn's correction.

### Mosquito infection with transgenic parasites

Transgenic parasites were cultured in the asexual blood and gametocyte stages and fed to mosquitoes as described above, with the following specifications for ATC usage. Both asexual and gametocyte cultures were maintained in 500 nM ATC (dissolved in DMSO). One hour prior to mosquito infection, ATC was washed out from the gametocyte cultures by replacing the media three times, and either 500 nM ATC or 0.025% DMSO was added to the blood meal. Mosquitoes were maintained on a 10% glucose solution supplemented with 100 μM ATC, which was dissolved directly in the solution.

### Culturing of primary human hepatocytes and infection with transgenic parasites

Cryopreserved human primary hepatocytes (BioIVT) were thawed, seeded on collagen-coated micropatterned islands in 96 well plates, and infected with WT or PfSIP2-cKD sporozoites as previously described[79]. In short, WT or PfSIP2-cKD-infected mosquito salivary glands were dissected in Schneider's medium (Gibco) containing 200 U ml$^{-1}$ of penicillin, 200 μg ml$^{-1}$ of streptomycin (2×, Gibco) for less than 1 h. Sporozoites were released from glands, filtered through a 35 μm cell strainer and counted on a haemocytometer. Parasites were spun at 10,000$g$ for 3 min, resuspended in hepatocyte media containing 2.5 μg ml$^{-1}$ Fungizone (Cytiva) and 2× penicillin/streptomycin as well as either 500 nM ATC or 0.025% of DMSO and seeded onto hepatocytes. For invasion assays, 7,000 sporozoites were seeded per well, wells were fixed in 4% paraformaldehyde 3 hpi and in and out PfCSP staining was performed[79]. For infection assays, 70,000–100,000 sporozoites were used, and subsequently washed off 3 hpi before supporting cells from the male mouse fibroblast cell line 3T3-J2 (ref. 80) were added. Cells were washed daily and wells were fixed at 2 dpi in 100% methanol and stained with rabbit PfHSP70 monoclonal primary antibody (antibodies-online, ABIN361730) and goat anti-rabbit IgG 546.

### Immunofluorescent microscopy

Midguts were dissected from female mosquitoes at specified time points. Any remaining blood bolus was removed in ice-cold PBS when present and clean midguts were incubated at room temperature in 4% paraformaldehyde for 45 min, followed by 3 washes in PBS for 10 min each. Midguts were permeabilized and blocked in 0.1% Triton X-100, 3% bovine serum albumin (BSA) in PBS for 1 h at room temperature or overnight at 4 °C before incubation in primary antibody at room temperature for 2 h, shaking. The following primary antibodies were used: 1:200 anti-*Toxoplasma gondii* centrin-1 rabbit polyclonal (Kerafast EBC004); 1:5 anti-*Drosophila* armadillo mouse monoclonal supernatant N2 7A1 (ref. 81) (deposited to the DSHB by E. Wieschaus), 1:200 anti-Pfs25 mouse monoclonal 4B7 (ref. 82) (deposited to BEI resources by L. H. Miller and A. Saul). Another 3 washes in PBS were performed before incubating the samples in secondary antibodies and phalloidin for 1 h at room temperature rocking (goat anti-rabbit IgG AlexaFluor 488, goat anti-mouse IgG AlexaFluor 488, goat anti-mouse AlexaFluor 568 from Invitrogen used at 1:400, Ebioscience Phalloidin eFluor 660 used at 1:300). Midguts were then stained in PBS containing 5 μg ml$^{-1}$ DAPI before the remaining 3 washes in PBS only. Midguts were mounted using Vectashield HardSet Antifade Mounting Medium and imaged on a Zeiss Inverted Observer Z1 with Apotome3 or a Zeiss LSM 980 confocal microscope. Using ZEN 3.10 software, confocal images were processed with LSM Plus whereas Airyscan images were processed with default parameters, followed by brightness and contrast adjustments in Fiji (v.2.14.0). Gamma was not adjusted unless specified.

### Statistical analysis

Specific statistical tests are detailed in the corresponding figure legend. Confirmation of EcR knockdown and analysis of parasite prevalence, intensity and infection rates were conducted using GraphPad Prism

10.1.1. Odds ratios of dual-read parasites in mosquito cell types were calculated in R using Fisher's exact tests followed by correction for multiple comparisons with Benjamini–Hochberg multiple comparison correction.

## Reporting summary

Further information on research design is available in the Nature Portfolio Reporting Summary linked to this article.

## Data availability

Reference genomes from *P. falciparum* and *A. gambiae* were obtained from PlasmoDB (https://plasmodb.org/plasmo/app) and VectorBase (https://vectorbase.org/vectorbase/app), respectively (release v.58)[54]. The published datasets were obtained from the Malaria Cell Atlas website (www.malariacellatlas.org) and from the Gene Expression Omnibus (GEO) repository, accession number GSE222586 (refs. 21,22). The source data for the single-cell analysis are accessible from the GEO repository, accession number GSE284537. Additional source data, including mosquito infection data, are deposited into Harvard Dataverse repository (https://doi.org/10.7910/DVN/DCNUFV). Source data are provided with this paper. Datasets generated in this study are archived on Zenodo (https://doi.org/10.5281/zenodo.16876946 (ref. 83)).

## Code availability

The scripts used for data processing and analysis are hosted on GitHub at https://github.com/lverzier/Pfalciparum_midgut_singlecell. Scripts generated in this study are archived on Zenodo (https://doi.org/10.5281/zenodo.16876946 (ref. 83)).

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

**Acknowledgements** The authors thank R. Dinglasan and M.-J. Gubbels for insights on the centrin antibody; J. D. Dvorin for the TetR-DOZI and guide plasmids; M. Delves for the NF54_H parasites; S. E. Lindner for valuable discussions on rRNA; K. W. Deitsch for insights into PfEMP1; members of the Catteruccia laboratory for comments; L. Ricci for help with graphics; and N. Damaraju for assistance with hepatocyte infections. The armadillo monoclonal antibody N2 7A1 developed by E. Wieschaus was obtained from the Developmental Studies Hybridoma Bank, created by the NICHD and maintained at The University of Iowa. We are deeply grateful to VEuPathDB for aiding the analysis of the datasets. F.C. is funded by the Howard Hughes Medical Institute (HHMI) as an Investigator and by the National Institutes of Health (NIH) grants R01AI148646 and R01AI153404. L.e.d.V. was funded by a Rubicon grant (452021309) from the Dutch Research Council (NWO).

**Author contributions** Y.Y., E.C., W.R.S., D.P. and F.C. conceived the study. Y.Y., E.C., E.D. and A.S.P. performed the dissections for the scRNA experiments, and Y.Y. and E.C. performed the single-cell isolation and scRNA library generation. Y.Y., L.H.V., and D.P. performed the scRNA analysis. Y.Y., L.H.V., E.C., F.A. and T.A.R. performed and analysed drug infection experiments. Y.Y., E.D. and L.e.d.V. generated transgenic parasites. Y.Y. performed PfSIP2-cKD experiments and Y.Y., L.H.V., S.M. and J.K. performed the primary human hepatocytes infections. E.N. reared mosquitoes used in the study. N.S. generated gametocyte culture and infected mosquitoes. L.H.V., E.C. and A.R.C. performed immunofluorescence imaging and analysis. S.N.B., D.P., W.R.S. and F.C. provided supervision and oversaw experiments and analyses. Y.Y., L.H.V., E.C. and F.C. wrote the original draft. Y.Y., L.H.V., W.R.S. and F.C. reviewed and edited the manuscript. All authors approved the final manuscript.

**Competing interests** The authors declare no competing interests.

**Additional information**
**Correspondence and requests for materials** should be addressed to Flaminia Catteruccia.

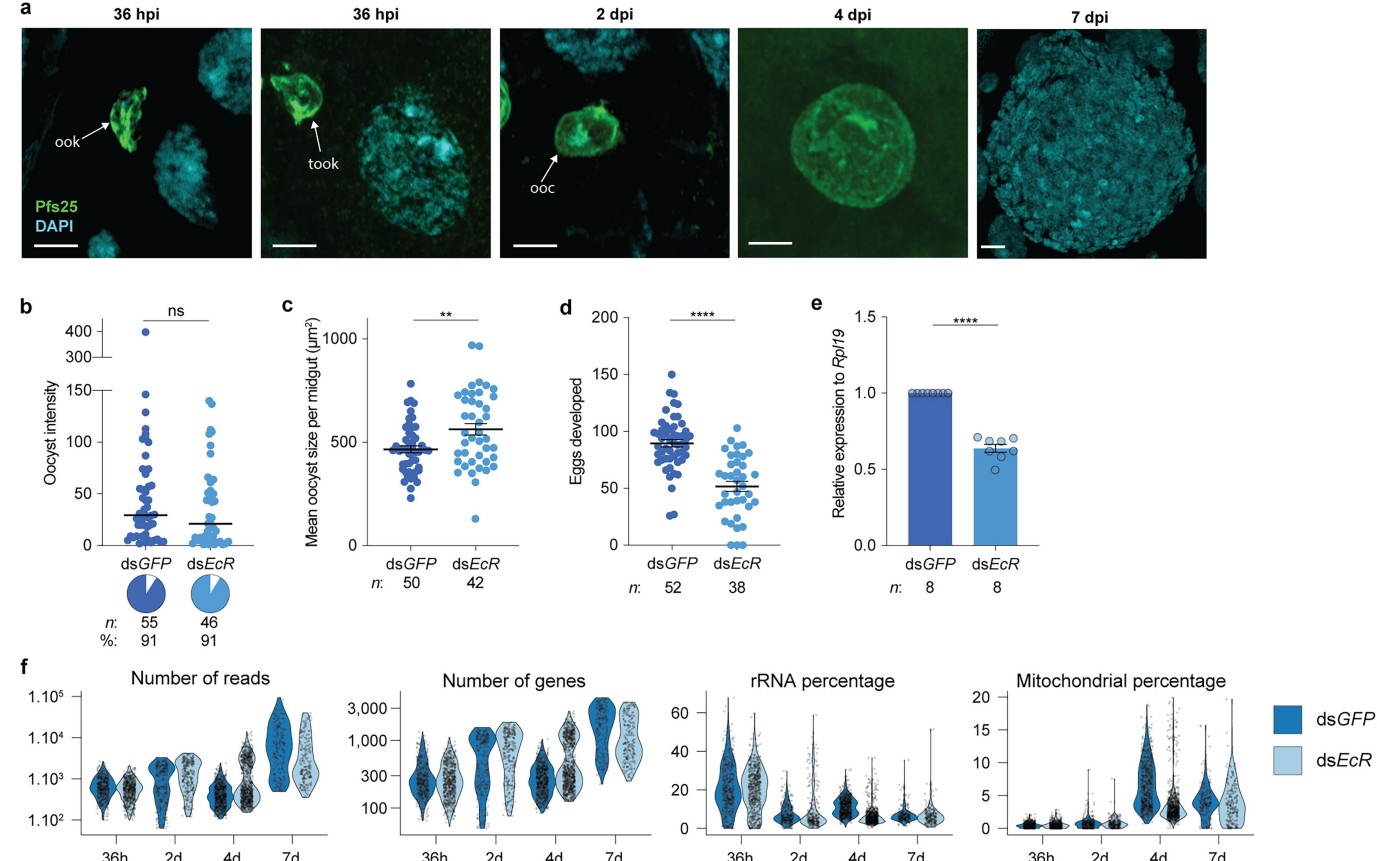

**Extended Data Fig. 1 | Quality control (QC) of scRNA-seq in two metabolic conditions. a**, Representative images of the parasite forms collected during the four experimental time points. Ookinetes (ook) and tooks found at 36 hpi, and oocysts (ooc) identified at 2 dpi, 4 dpi, and 7 dpi. Parasites were stained with Pfs25 (green) and nuclei with DAPI (cyan). Scale bar: 5 μm. **b-c**, *EcR* knockdown across four replicates (**b**) did not affect oocyst prevalence (pie charts, Fisher's exact test, two-tailed) or intensity (Mann Whitney test, two-tailed), but

(**c**) increased oocyst size at 7 dpi (Mann Whitney test, two-tailed). **d-e**, Knockdown of *EcR* was confirmed by (**d**) detecting the expected reduction in egg numbers in the ovaries (unpaired t-test, two-tailed) and (**e**) RT-qPCR (unpaired t-test, two-tailed). **f**, Violin plots displaying key metrics of the *P. falciparum* scRNA-seq data after QC. *n* indicates the number of mosquitoes, % the prevalence of infected mosquitoes. Data are represented as mean ± SEM in panels **c**, **d** and **e**, and as median in panels **b**.

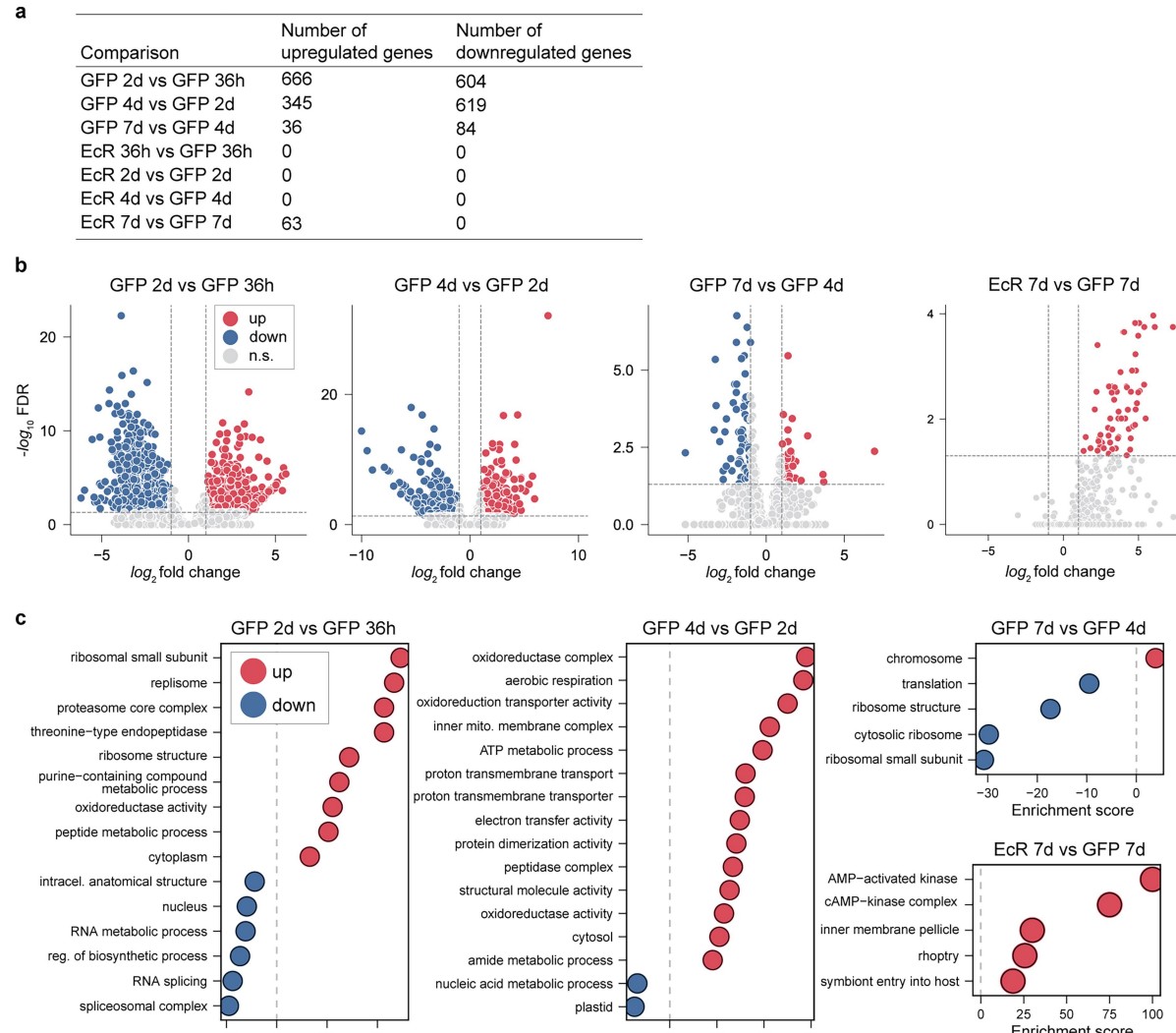

**a**

| Comparison | Number of upregulated genes | Number of downregulated genes |
|---|---|---|
| GFP 2d vs GFP 36h | 666 | 604 |
| GFP 4d vs GFP 2d | 345 | 619 |
| GFP 7d vs GFP 4d | 36 | 84 |
| EcR 36h vs GFP 36h | 0 | 0 |
| EcR 2d vs GFP 2d | 0 | 0 |
| EcR 4d vs GFP 4d | 0 | 0 |
| EcR 7d vs GFP 7d | 63 | 0 |

**Extended Data Fig. 2 | Pseudobulk differential expression analysis of *P. falciparum* across time and treatment. a**, Table summarizing the number of upregulated and downregulated genes identified in the comparisons of time points and/or treatments by pseudobulk (significance defined as FDR < 0.05 and absolute value of $\log_2$ fold change >1). **b**, Volcano plots representing the differentially regulated genes in the four comparisons that yield significant changes. **c**, Corresponding GO terms that are significantly enriched in upregulated and downregulated genes (Fisher's one-tailed test, g:SCS algorithm for multiple test correction).

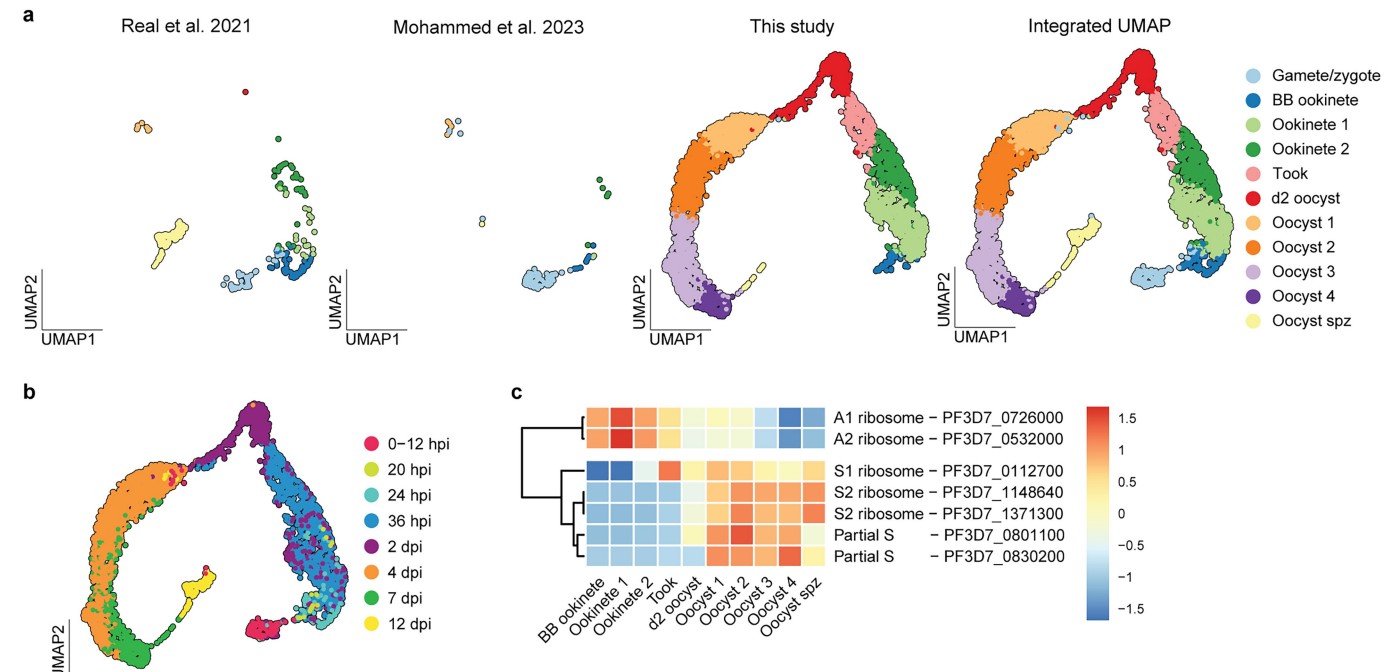

**Extended Data Fig. 3 | Integrated *P. falciparum* UMAPs across data origin and time. a-b**, UMAP plots representing the integrated data of *P. falciparum* (**a**) split by their study of origin and (**b**) coloured by time points of single cell collection. **c**, Heatmap showing the Z scores of expression patterns of ribosomal RNA (rRNA). Each row represents a cumulative expression of 18S (when present), 5.8S and 28S for each rRNA type (A1, A2, S1, S2). The gene ID given is of the corresponding 28S.

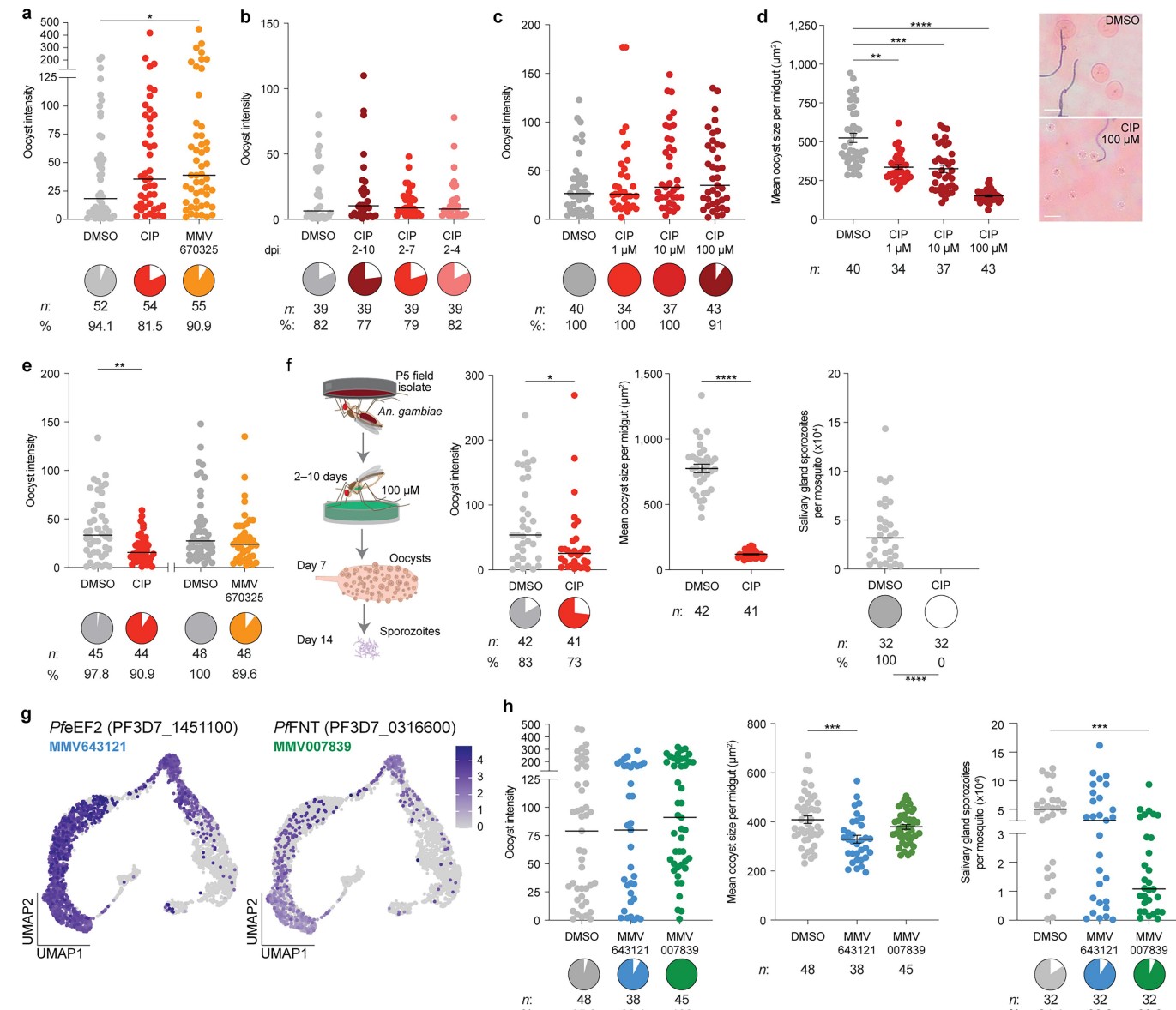

**Extended Data Fig. 4 | Functional validation of candidate genes in oocyst growth and sporozoite infection. a**, CIP and MMV670325 delivered in sugar from 2–10 dpi to *An. gambiae* infected with NF54 had no effect on oocyst prevalence, while MMV670325 treatment slightly increased oocyst intensity (p = 0.041). **b-d**, CIP exposure for three different **(b)** durations or **(c-d)** concentrations had no effect on **(b-c)** oocyst prevalence or intensity but **(d)** significantly reduced oocyst size in a dose dependent manner (p = 0.0033, p = 0.0002, p < 0.0001; representative images of DMSO and 100 μM CIP treatment in right panel, Scale bar: 20 μm). **e**, The same treatments to *An. stephensi* infected with the ART29 isolate resulted in no prevalence difference, but a slight decrease in oocyst intensity (p = 0.002) after CIP treatment. **f**, CIP treatment in *An. gambiae* infected with the P5 isolate slightly decreased oocyst numbers (p = 0.016), but significantly reduced oocyst size (p < 0.0001) at 7 dpi, resulting in no

sporozoites in the salivary glands at 14 dpi. **g**, Expression profile of *Pfe*EF2 and *Pf*FNT, targets of MMV643121 and MMV007839, respectively. **h**, Exposure to either drug did not affect oocyst prevalence or intensity. MMV643121 treatment led to a decrease in oocyst size (p = 0.0005), while MMV007839 did not affect oocyst size but resulted in reduced sporozoite intensity (p = 0.0009). Oocyst size data are represented as mean ± SEM, oocyst and sporozoite intensity data as medians. All are compared using either a Mann Whitney two-tailed test (panels **e** and **f**) or a Kruskal-Wallis test and Dunn's correction (all other panels). *n* indicates the number of mosquitoes from at least two independent infections. Pie charts indicate infection prevalences, compared with two-tailed Fisher's exact test. *p < 0.05, **p < 0.01, ***p < 0.001, ****p < 0.0001. Schematics in **f** are adapted from ref. 36, CC BY 4.0.

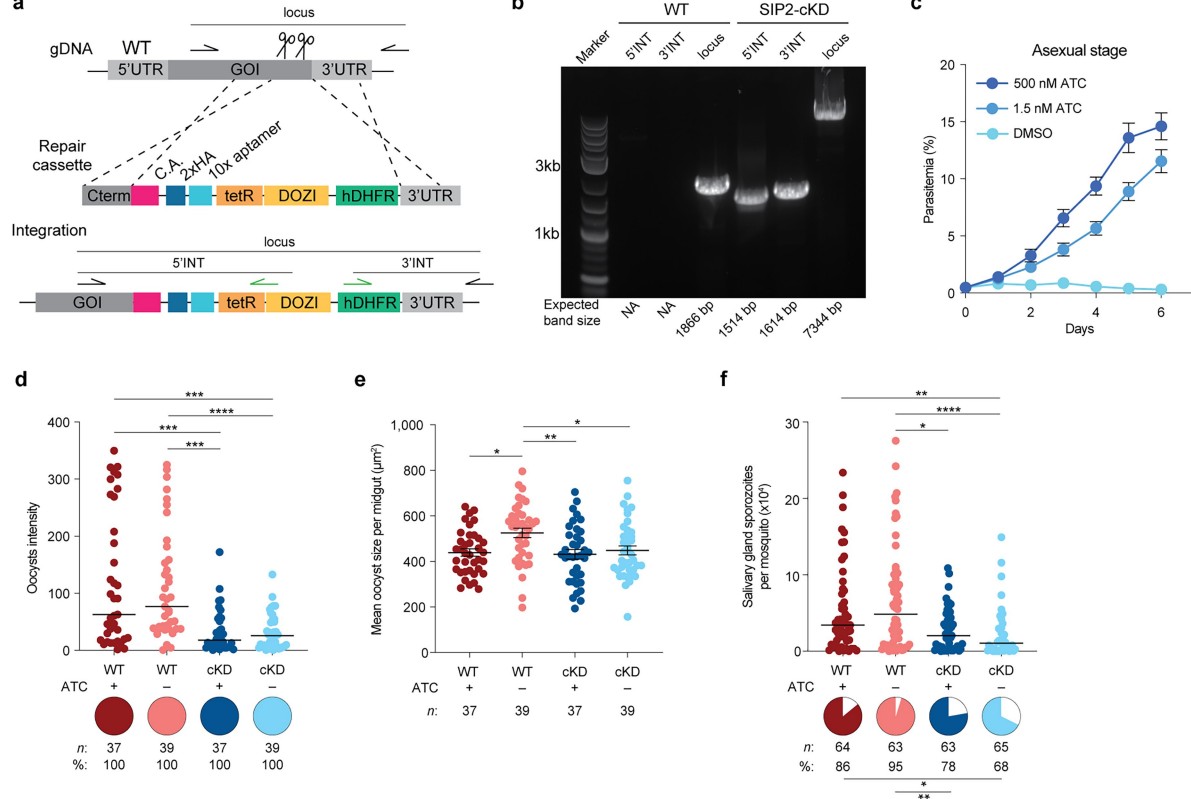

**Extended Data Fig. 5 | Conditional knockdown of PfSIP2 does not affect sporozoite intensity. a**, Scheme (not to scale) of the *Pf*SIP2 conditional knockdown (cKD) construct. Positions of primer pairs for integration validation are indicated. **b**, PCR amplifications of the 5′ and 3′ integrated regions (5′INT and 3′INT) and locus in WT NF54 and *Pf*SIP2-cKD parasites, demonstrating the absence of WT parasites in the cKD line. For gel source data, see Supplementary Fig. 1. **c**, *Pf*SIP2 knockdown by ATC withdrawal in the asexual blood stage led to parasite death (Mixed-effects model, Dunn's correction, p < 0.0001). **d-f**, PfSIP2 knockdown by ATC withdrawal had no effect on (**d**) oocyst prevalence (Fisher's exact test, two-tailed), intensity

(Kruskal-Wallis test, Dunn's correction, p = 0.0004, p = 0.0007, and p < 0.0001), and (**e**) size (Kruskal-Wallis test, Dunn's correction, p = 0.0152, p = 0.0082, and p = 0.0258) at 7 dpi, and had no effect on (**f**) prevalence (p = 0.0211, p = 0.0076, and p < 0.0001) and intensity (p = 0.0024, p = 0.0213, and p < 0.0001) of salivary gland sporozoites at 14 dpi (two-tailed Fisher's exact tests and Kruskal-Wallis test with Dunn's correction, respectively). Data are represented as mean ± SEM in panels **c**, and **e**, and as median in **d**, and **f**. *n* indicates the number of mosquitoes and pie charts and % infection prevalence. *p < 0.05, **p < 0.01, ***p < 0.001, ****p < 0.001.

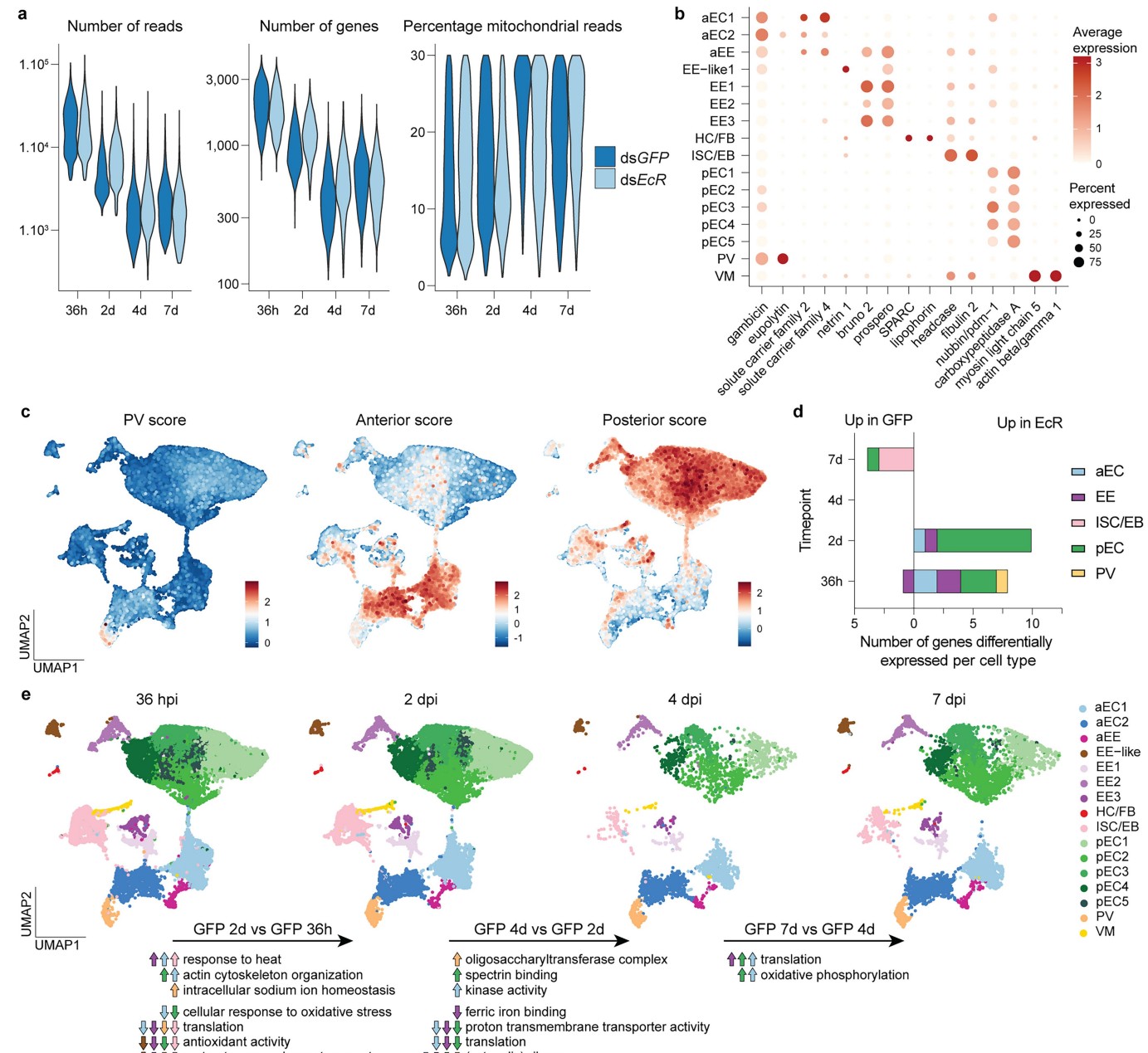

**Extended Data Fig. 6 | Quality control and clustering of the *An. gambiae* scRNA-seq data. a**, Violin plots displaying key quality metrics for the mosquito scRNA-seq datasets (left to right: total number of reads, of genes, and percentage of mitochondrial reads) after QC. **b**, Dot plot displays cell type-specific markers, showing their average expression (colour) and the proportion of expressing each marker (size). Cell types shown are anterior enterocytes (aEC), enteroendocrine (EE), EE-like cluster, haemocytes and fat body cell (HC/FB), intestinal stem cells/enteroblasts, (ISC/EB), posterior enterocytes (pEC), proventriculus or cardia (PV) and visceral muscles (VM). **c**, Expression of markers characteristic to the proventriculus, and anterior or posterior part of the midgut based on the study by Hixson et al.[69]. **d**, Number of genes differentially expressed by pseudobulk analysis between ds*GFP*- and ds*EcR*-treated mosquitoes per time point and cell type (Wald test, significance defined as adjusted p-value < 0.05 and absolute value of log$_2$ fold change > 0.5). **e**, UMAPs of *An. gambiae* cells split by time point, showing the dynamic changes of the midgut cell population shortly after blood digestion (36 hpi and 2 dpi) and when subsequently maintained on sugar-only diet (4 dpi and 7 dpi). Pseudobulk differential expression analysis was performed in ds*GFP*-treated mosquitoes by pooling each cell type and comparing consecutive time points. Changes to significant GO term enrichment are represented by up and down arrows, coloured by cell type.

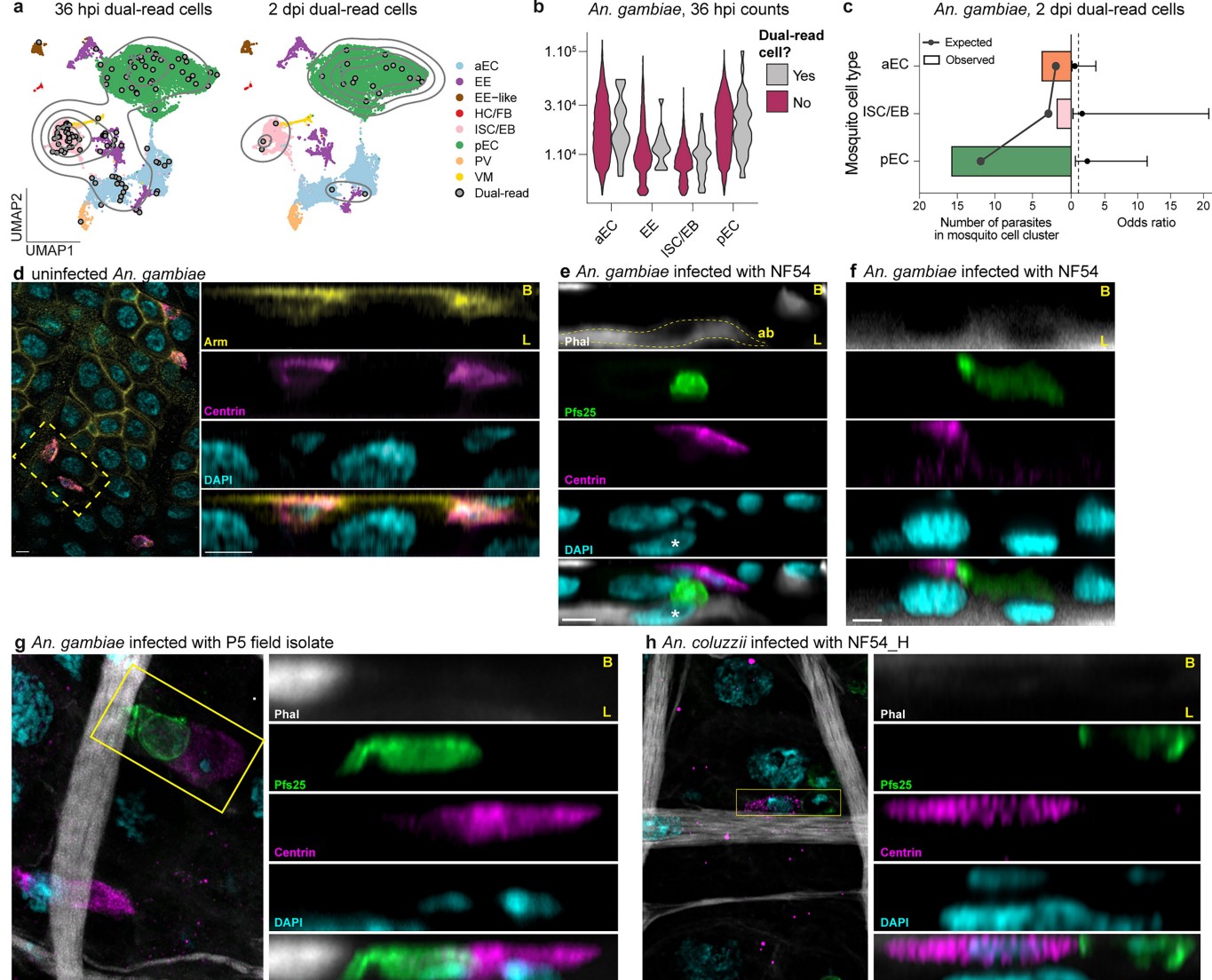

**Extended Data Fig. 7 | Dual-read parasites at 36 hpi show an association with intestinal progenitor cells. a**, *An. gambiae* UMAPs at 36 hpi and 2 dpi coloured by cell type and overlaid with dual-read cells (gray circles) and contour lines (light gray lines) displaying the density of dual-read cells. **b**, Violin plot showing similar read counts in mosquito cells regardless of whether they are dual-read cells. **c**, twenty-one dual-read parasites over eight independent samples at 2 dpi were randomly distributed across mosquito cell types, with no differences in (left x-axis) the observed vs. expected numbers and (right x-axis) resulting odds ratio (point) ± 95% confidence interval (error bar). Fisher's exact test, two-tailed, with BH multiple comparison correction. **d**, Maximum intensity projection from confocal imaging (9.66 μm Z-stack) showing colocalization of the centrin (magenta) signal with the well-characterized ISC marker, armadillo (Arm, yellow), located on the basal side of the midgut (B, top). **e-f**, Confocal imaging of NF54 parasites (Pfs25, green) traversing the *An. gambiae* midgut epithelium at 36 hpi. (**e**) Parasites were found to point towards a progenitor cell (centrin, magenta) directly above an extruding cell (asterisk, nucleus of the extruded cell), located between two actin brushes (ab, yellow dashed lines) delimiting each epithelium (phalloidin, gray, gamma 0.5). (**f**) Ookinete left a Pfs25 trail into a large cell reminiscent of an enterocyte (see Supplementary Video 1). **g-h**, The interaction was also observed in (**g**) *An. gambiae* midguts infected with P5 parasites and (**h**) *An. coluzzii* midguts infected with NF54_H parasites (max intensity projection of 1.69 and 1.68 μm Z-stacks, respectively). Imaging was performed on at least two independent infections. Phalloidin (phal, gray) was used to define the basal side (B, top) and the luminal brush border (L, bottom), and nuclei were stained with DAPI (cyan). Scale bar: 5 μm.

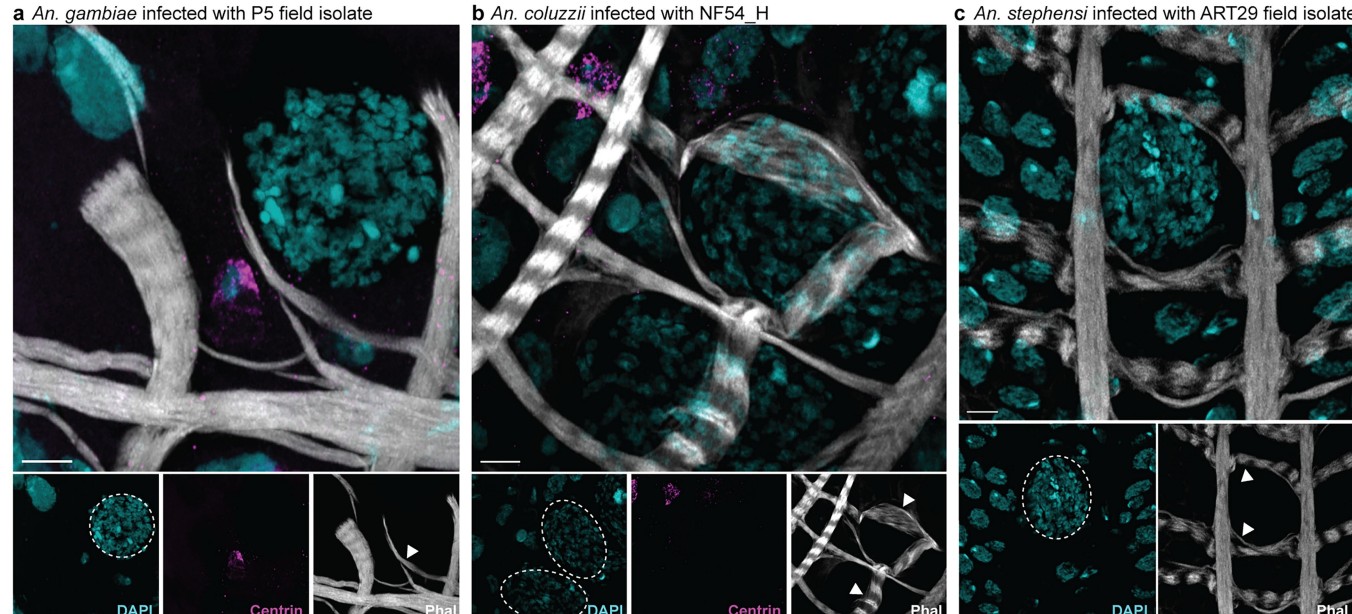

**a** *An. gambiae* infected with P5 field isolate

**b** *An. coluzzii* infected with NF54_H

**c** *An. stephensi* infected with ART29 field isolate

**Extended Data Fig. 8 | Visceral muscle wraps around late-stage oocysts.**
**a-b**, Visceral muscle (Phal, gray) wrapped closely around late oocysts (arrowheads), with no apparent increase in progenitor cell (centrin, magenta) populations near oocysts in (**a**) *An. gambiae* midguts infected with P5 parasites and (**b**) *An. coluzzii* midguts infected with NF54_H parasites by confocal imaging (max intensity projection of 4.60 μm and 2.60 μm Z-stacks, respectively). **c**, Similar muscle deformation was observed in *An. stephensi* midguts infected with ART29 parasites (max intensity projection of 3.60 μm Z-stack), though inconsistent centrin staining prevented analysis of progenitor cell distribution. At least two independent infections were used for imaging. Scale bar: 5 μm.

# Reporting Summary

## Statistics

For all statistical analyses, confirm that the following items are present in the figure legend, table legend, main text, or Methods section.

| n/a | Confirmed | |
|---|---|---|
| ☐ | ☒ | The exact sample size (*n*) for each experimental group/condition, given as a discrete number and unit of measurement |
| ☐ | ☒ | A statement on whether measurements were taken from distinct samples or whether the same sample was measured repeatedly |
| ☐ | ☒ | The statistical test(s) used AND whether they are one- or two-sided *Only common tests should be described solely by name; describe more complex techniques in the Methods section.* |
| ☐ | ☒ | A description of all covariates tested |
| ☐ | ☒ | A description of any assumptions or corrections, such as tests of normality and adjustment for multiple comparisons |
| ☐ | ☒ | A full description of the statistical parameters including central tendency (e.g. means) or other basic estimates (e.g. regression coefficient) AND variation (e.g. standard deviation) or associated estimates of uncertainty (e.g. confidence intervals) |
| ☐ | ☒ | For null hypothesis testing, the test statistic (e.g. *F*, *t*, *r*) with confidence intervals, effect sizes, degrees of freedom and *P* value noted *Give P values as exact values whenever suitable.* |
| ☒ | ☐ | For Bayesian analysis, information on the choice of priors and Markov chain Monte Carlo settings |
| ☒ | ☐ | For hierarchical and complex designs, identification of the appropriate level for tests and full reporting of outcomes |
| ☒ | ☐ | Estimates of effect sizes (e.g. Cohen's *d*, Pearson's *r*), indicating how they were calculated |

*Our web collection on statistics for biologists contains articles on many of the points above.*

## Software and code

Policy information about availability of computer code

| Data collection | Our single cell library preparation was tested on the Illumina iSeq100 before running on two NovaSeq S4 flow cells for deep sequencing. |
|---|---|
| Data analysis | Single-cell analysis pipeline is described in the Code Repository. All other analysis can be found in the Methods section of the manuscript. The following software are used in this study: 10X Cell Ranger software v7.0.1, Python v3.10, Scanpy v1.9.1, Scrublet v0.2.3, SeuratDisk (v0.9021), Seurat (v5.0.2), Pegasus (v1.8.1) bioconductor-deseq2 (v1.42.0), GoProfiler (https://biit.cs.ut.ee/gprofiler/gost), R (v4.3.2), RStudio (v2023.9.1.494), Monocle3 (v1.3.4), scDblFinder(v1.16.0), OocystMeter (https://www.biorxiv.org/content/10.1101/2025.06.28.662088v1), ZEN (v3.10), Fiji (v2.14.0), GraphPad Prism (v10.1.1). |

For manuscripts utilizing custom algorithms or software that are central to the research but not yet described in published literature, software must be made available to editors and reviewers. We strongly encourage code deposition in a community repository (e.g. GitHub). See the Nature Portfolio guidelines for submitting code & software for further information.

## Data

Policy information about availability of data

All manuscripts must include a data availability statement. This statement should provide the following information, where applicable:
- Accession codes, unique identifiers, or web links for publicly available datasets
- A description of any restrictions on data availability
- For clinical datasets or third party data, please ensure that the statement adheres to our policy

Reference genomes from P. falciparum and An. gambiae were obtained from PlasmoDB (https://plasmodb.org/plasmo/app) and VectorBase (https://vectorbase.org/vectorbase/app) respectively. The published datasets were obtained from the Malaria Cell Atlas website (www.malariacellatlas.org) and from the GEO repository, accession number GSE222586. The source data for the single cell analysis are accessible from the GEO repository, accession number GSE284537. Additional source data, including mosquito infection data, are deposited into Harvard Dataverse repository (https://doi.org/10.7910/DVN/DCNUFV).

## Research involving human participants, their data, or biological material

Policy information about studies with human participants or human data. See also policy information about sex, gender (identity/presentation), and sexual orientation and race, ethnicity and racism.

| | |
|---|---|
| Reporting on sex and gender | N/A: Donors were anonymized so sex and gender are not known. |
| Reporting on race, ethnicity, or other socially relevant groupings | N/A: Donors were anonymized so ethnicity is not known. |
| Population characteristics | N/A: Donors were anonymized so any population characteristics are not known. |
| Recruitment | N/A: No recruitment is involved in the study. |
| Ethics oversight | N/A |

Note that full information on the approval of the study protocol must also be provided in the manuscript.

# Field-specific reporting

Please select the one below that is the best fit for your research. If you are not sure, read the appropriate sections before making your selection.

☒ Life sciences          ☐ Behavioural & social sciences          ☐ Ecological, evolutionary & environmental sciences

For a reference copy of the document with all sections, see nature.com/documents/nr-reporting-summary-flat.pdf

# Life sciences study design

All studies must disclose on these points even when the disclosure is negative.

| | |
|---|---|
| Sample size | We performed four independent biological replicates for our single cell data collection to collect enough parasites, due to the difference in number of parasites compared to mosquito cells (Graumans et al 2020 Trends in parasitology, PMID: 32620501), and account for biological variation inherent to an infection model. Parasite growth assays and mosquito infection experiments were all performed in triplicate, unless specified, as is the norm in the field (Probst et al. 2025 Nature, PMID: 40399670). |
| Data exclusions | No data were excluded. |
| Replication | 2-4 biological replicates were performed and all replicate attempts were successful. |
| Randomization | Mosquitoes were collected as pupae and randomly assigned to different cages representing distinct treatment groups. Mosquitoes were then randomly aspirated from cages for double-stranded RNA injection, dissection for single-cell isolation, oocyst and sporozoite quantification, or microscopy analysis. For the asexual-stage growth assay, PfSIP2-cKD parasites were randomly assigned to wells containing different drugs. |
| Blinding | Blinding was not necessary as all mosquito infections were analyzed using the software OocystMeter. |

# Reporting for specific materials, systems and methods

We require information from authors about some types of materials, experimental systems and methods used in many studies. Here, indicate whether each material, system or method listed is relevant to your study. If you are not sure if a list item applies to your research, read the appropriate section before selecting a response.

## Materials & experimental systems

| n/a | Involved in the study |
|---|---|
| ☐ | ☒ Antibodies |
| ☐ | ☒ Eukaryotic cell lines |
| ☒ | ☐ Palaeontology and archaeology |
| ☐ | ☒ Animals and other organisms |
| ☒ | ☐ Clinical data |
| ☒ | ☐ Dual use research of concern |
| ☒ | ☐ Plants |

## Methods

| n/a | Involved in the study |
|---|---|
| ☒ | ☐ ChIP-seq |
| ☒ | ☐ Flow cytometry |
| ☒ | ☐ MRI-based neuroimaging |

## Antibodies

| | |
|---|---|
| Antibodies used | Centrin-1 antibody, Kerafast, EBC004; Armadillo antibody, DSHB, N27A1; pfs25 antibody, BEI, MRA28, Clone 4B7, Lot#70013640; Phalloidin eFluor 660, Ebioscience, 50-6559-05; PfHSP70, antibodies-online, ABIN361730; goat anti-rabbit IgG 546, Invitrogen, A-11035; goat anti-rabbit IgG AlexaFluor 488, Invitrogen, A-11008; goat anti-mouse IgG AlexaFluor 488, Invitrogen, A-11001; goat anti-mouse AlexaFluor 568, Invitrogen, A-11004. DAPI, Millipore Sigma, D9542-5MG. |
| Validation | Centrin-1 antibody validation via co-staining with Armadillo for specificity staining progenitor cells. See Dinglasan et al. PNAS. 2007 for pfs25 staining. Spradling and Ohlstein. Nature. 2005 for armadillo staining. Chazotte. Cold Spring Harb Protoc. 2010. for phalloidin. Goel et al. Nature medicine 2015 for HSP70. |

## Eukaryotic cell lines

Policy information about cell lines and Sex and Gender in Research

| | |
|---|---|
| Cell line source(s) | NF54 P. falciparum cell line from BEI Resources (MRA-1000), and field isolates ART29 and P5 (Paton, D. G. et al. 2022 PLoS Pathog) |
| Authentication | Parasite line was authenticated using nested PCR protocol with primers specific to P. falciparum. |
| Mycoplasma contamination | All P. falciparum strains were confirmed to be free of mycoplasma contamination. |
| Commonly misidentified lines (See ICLAC register) | *Name any commonly misidentified cell lines used in the study and provide a rationale for their use.* |

## Animals and other research organisms

Policy information about studies involving animals; ARRIVE guidelines recommended for reporting animal research, and Sex and Gender in Research

| | |
|---|---|
| Laboratory animals | Anopheles gambiae (G3 strain), An. coluzzii dervied from the field (Adams, K. L. et al. 2023 PLoS Pathog), and An. stephensi SDA-500. One-day-old female mosquitoes were used for double-stranded RNA injection, and 4-7-day-old female mosquitoes were used for other experiments. |
| Wild animals | No wild animals were used in the study. |
| Reporting on sex | Only female mosquitoes used, as males do not feed on blood and therefore do not carry P. falciparum. |
| Field-collected samples | No field collected samples were used in the study. |
| Ethics oversight | Lower invertebrates, such as mosquitoes, are exempt from regulations governing the use of animals in research. |

Note that full information on the approval of the study protocol must also be provided in the manuscript.

## Plants

Seed stocks

N/A

Novel plant genotypes

N/A

Authentication

N/A

