## [Peer Review file · Nature]

Mapping Plasmodium transitions and interactions in the Anopheles female

Corresponding Author: Flaminia Catteruccia

Version 0:

Reviewer comments:

Referee #1

(Remarks to the Author)

In Yan et al., the authors assert that they have advanced our knowledge (a "momentous step", line 304) of malaria parasite transitions at the single cell RNA and cellular scales in *Anopheles gambiae*. If we accept the description of gaps in knowledge in the background, which lacks acknowledgement of previous work in some uncomfortable ways, there is an argument here. However, the data presented are more accurately contextualized as bridging some gaps in previous work that should be better acknowledged. Further, quite a few findings are largely unsurprising, with upregulated genes for cytoskeletal transitions during midgut invasion and increased aerobic respiration with oocyst growth, for example, with only a few novel findings (e.g., PfSIP2-dependent invasion of hepatocytes). As an aside, the latter is awkwardly included, given the focus and title on parasite development in the mosquito host. Drugs (CIP, MMV5) are tested at very high single concentrations (100uM); the choice of this dose is not explained, and effects should be presented over a range of doses for this first use of these drugs in mosquitoes. The authors point out in the discussion that *P. falciparum* is distinct in its biology from *P. berghei*. While this is obviously true, the work presented is focused on a single *P. falciparum* strain in a single strain of *A. gambiae*. What we really need to know is how variable such findings are across different *P. falciparum* strains in different strains of *A. gambiae*, perhaps even in *A. coluzzii* and in *A. stephensi*. Such comparisons would reveal how fundamental and applicable these findings are that the authors state are already plausible targets for transmission blocking. Without data on other *falciparum* strain/mosquito strain and species combinations, we lack a critical understanding as to whether these findings are unique to this parasite-mosquito combination or truly biologically fundamental for publication at the hoped for level. Beyond the authors surprising assertion that "How surviving ookinetes interact with midgut cells and form oocysts is unknown," which is more accurately described as "poorly" known, there are other examples of inaccurate statements. In lines 325-327, the authors write that "Recent studies suggest that..." with citations 4, 12 and 42. Citation 12 is from 2005, citation 4 is from 2007 and citation 42 is the only recent publication from 2023. This is picky, yes of course, but it fits unfortunately with the impression of there being more than a handful of over-statements and inaccurate statements in the text. Of note from their mosquito biology data, the authors need to clearly cite previously published, definitive work that has confirmed the existence and functional biology of distinct *A. gambiae* midgut cell types (e.g., ISCs/EBs, pECs, aECs, etc). These descriptions have become established through copied inferences from a few papers rather than a citation of works describing functionally validated, definitive cell types in *A. gambiae* or anophelines in general. I know of no such work that has definitively and functionally identified these midgut cell types in *A. gambiae*, but hopefully the authors can clarify this in a revision. Overall, I would judge this work as a very good, incremental addition to the literature for a more focused audience, but there is also work to be done to address over-interpretations and inaccurate statements, an uncomfortable lack of acknowledged previous work (even with the citation limitation this is entirely possible), and some writing challenges that detract from readability. In addition, there needs to be at least partial confirmation of major observations using multiple combinations of *falciparum* strains and *A. gambiae* strains/other major *Anopheles* vector species to deliver us to an understanding of biology that is truly fundamental to *P. falciparum* invasion and development within *Anopheles* spp.

Referee #2

(Remarks to the Author)

This is comprehensive study that uses scRNAseq well, identifying patterns, and then testing the potential importance of

some genes of interest in human malaria parasites. The work targeting some follow up genes is solid and convincing. A bit more detail on whether other genes were also tested and found to have no effect, as well as how these genes in particular were decided on for functional assays would be welcome.

The authors' study of parasite interactions with mosquito cells gives nice resolution about interactions at these critical time points. While the results look fairly clean, the mosquito data are not shown to have been fully QC'd and this should be remedied to bring the manuscript to the standard of a Nature paper.

Appreciating that the details must be reserved for the Methods section, it would be beneficial to the reader to add in a sentence or two to the main text discuss how you isolated parasites and how you are sure in this process that the mosquito cells remained sufficiently intact to warrant joint scRNAseq analysis. Similarly, how you did QC and how many mosquito cells were assayed deserve a brief mention in the main text.

In Methods, please be clearer on how many 10x runs were completed and how you addressed batch effects if each treatment/time point was on a separate run. Provide details on cells loaded vs recovered for each run. Also, there is no mention of doublet removal – how did you address doublets?

In Methods, please expand on the work you did to confirm that the dissociation protocol led to viable parasite and mosquito cells. There is an early reference to "meticulously optimizing the protocol" and to see Methods for more details but these details are not provided in Methods. It is therefore unclear how you enriched for parasites or what optimizations were carried out. Some further expansion is warranted to showcase this optimization and bring more credibility to the protocol that results in good parasite AND mosquito cells (noting as below, this should also be published as a protocol). Did you do any staining for live/dead cells? If the protocol you use has been already shown to lead to viable epithelial tissue single cell suspensions in other studies, please cite the work that showed that. In Methods, you state that mitochondrial percentage per cell was used in quality control only for Plasmodium. This should be done for Anopheles as well. Why wasn't it done? Epithelial tissue is not straightforward to dissociate (hence many studies resort to single nuclei RNAseq) and the protocol, if it is demonstrated to lead to high quality mosquito gut cells, should be published in detail.

More information on the Anopheles cells should be presented in the main text before the "parasite association with midgut cells" section. This includes for example how many cells/runs/ were used and also please provide justification of the merging of Anopheles cells from both control and dsEcR as soon as the Anopheles cells start to be discussed (if indeed that is what was done). It's otherwise confusing to the reader. Cell type annotations for the Anopheles data are also not explained in enough detail. The manuscript only states "annotated manually by cross-referencing cluster markers from four references". Please provide more information on how this was done – e.g. what did you trust as enough concordance to call something a particular cell type?

Minor comments

Main text references to EDF 6c and d should be swapped.

Fig 1b, please put percent of variance explained by each PC.

Please comment on the typical size of a 7-day old oocyst and compatibility with any cell size limits of 10x. Presumably this limitation is why some older oocysts were not explored using 10x, but if it was tried and failed, it would be good to report here.

The last sentence of the discussion could be removed, it doesn't add a lot of value and this was already clear.

Referee #3

(Remarks to the Author)

This elegant study presents significant progress by taking our knowledge of parasite-vector interactions in malaria beyond the state of the art. Importantly, it does so not in a tractable rodent model but by looking at the more challenging and most important human malaria parasite, *P. falciparum*. For the first time, this study exploits the power of single cell transcriptomics to analyse the interacting cells of the parasite and vector midgut simultaneously. The combined analysis of parasite and midgut cellular transcriptomes produces an atlas of parasite development in the mosquito that is more highly resolved than previous studies in *P. berghei* or *P. falciparum*, thereby generating an essential new community resource. New clusters of co-regulated parasite genes are identified which can be predicted to enable a deeper understanding of parasite biology. The study additionally characterises cell types of the mosquito midgut at unprecedented depth. Conceptually, the use of single cell transcriptomics to explore the complexity of a parasite vector system is novel and exciting.

Examples are provided as proof of principle for how the new data can be exploited in future to develop transmission blocking interventions that target oocyst proteins and for gaining a deeper understanding of parasite biology. Specific biological insights in this manuscript do not provide complete, new mechanisms but provide inspiration and starting points for future research. I am convinced the data will inspire future discoveries. Illustrated starting points include potential targets for blocking sporogony with inhibitors delivered to mosquitoes, the role of a DNA binding protein for the liver stage and the interactions of different parasite stages with host cell types. All these experiments are well performed and push beyond the limits of current technologies, for instance by adapting the conditional gene knockdown system to study oocyst biology and sporozoite infectivity. That oocysts anchor themselves to the midgut using PfEMP1 adhesins and may upregulate muscle cells adhesive genes for the same purpose are provocative ideas that will be interesting to explore in future. The study provides a valuable resource for data, for new ideas, hypotheses and starting points for future work. I consider it to be a landmark study for the field.

The main value of the work is in the overall picture it paints of cell types of vector and parasite and their potential interactions. Individual validation experiments are not overinterpreted, with the exception of the minor points raised below. Deeper mechanistic studies, like visualising or knocking out parasite and muscle cell adhesins, would be possible but could be argued to be beyond the scope of this work, since it would not be necessary to increase my confidence in the utility of the resource and the validity of the overall picture it paint.

Overall, in my opinion the quality of the data is good, their analysis is appropriate, the presentation is clear and the conclusions are supported well by the data.

Minor points:

- Regarding the speculation that upregulated PfEMP1 proteins many anchor oocysts to midgut cells or basal lamina, are these genes uniquely upregulated in oocysts or also (on other available data sets) in asexual blood stages? Do they differ regarding their targeting signals, given that blood stages PfEMP1 proteins are targeted to the host cell plasma membrane?
- Did the authors find progenitor transcriptomes to be influenced by the presence of a parasite?
- Did mosquito cell transcriptomes change with time after feeding or EcR knock down?
- If there is a temporal component, were the transcriptomes of parasite-associated muscle cells compared with un-associated cells from the same sample, or could there be a temporal or batch effect explaining the 24 upregulated genes?
- Given that different size cell strainers were used, could the abundance of Oocyst 4-muscle doublets be the result of selective filtering of the day 7 samples? Are VM cells recovered equally at different time points, i.e. could earlier samples have physically removed muscle cell associated parasites through the strainer with smaller pore size?
- p. 8: They should tone down the conclusion that their data show PfSIP2 expression in late oocysts to be essential for the ability of sporozoites to infect hepatocytes. Although I agree this is likely, it isn't the only possibility. Gene expression in the sporozoite could be important even if below the single cell detection limit, or the protein could act in the sporozoite even though the transcript originates in the schizont. The highest expression level of a gene does not necessarily coincide with the most essential function, and that the sporozoite has some low but important level of transcription is impossible to rule out. Perhaps say "suggest" or include a "most likely".
- p. 9: The suggestion that VM cells respond to the presence of parasites by enhancing their adhesive properties is not the only explanation for the data. The statement should be qualified or spatial/imaging data should be provided that muscle fibres next to cysts express different proteins. Localisation of such proteins by microscopy would be exciting, of course. The single cell data alone may be explained differently. For instance, cells with more adhesins may simply be more likely to remain associated with and oocyst.
- Please, explain where the reported effects on splicing can be found in the data.

Version 1:

Reviewer comments:

Referee #2

(Remarks to the Author)

The authors have done a highly commendable job responding to all reviews.

I still have some concerns about the quality of the included mosquito single cell data and believe this should be commented on through additional text in the manuscript (not through additional experiments). The authors have now added in more details on mosquito cell QC and annotation. They elect to include mosquito cells up to 30% mitochondrial reads, but this is higher than I have ever heard of and there seems to be a pattern of just citing other papers that have used seemingly high and arbitrary thresholds even among the five papers the authors now cite on line 779. As far as I could tell, only one of the 5 cited papers went as high at 30%. The Fly Cell Atlas paper (which uses nuclei so is going to use lower thresholds) has its liberal setting at 15% and its strict setting at 5% mitochondrial reads. The authors have now included figure S6a that shows the distribution of the mitochondrial read percentage and it is striking that days 4 and 7 actually have the majority of cells showing well above 10% mitochondrial reads while this is not true for 36 hours, where most cells are more "normal". I would appreciate some attention to this in the manuscript text. Ultimately, these data are what they are and the authors need to make semi-arbitrary cut offs to carry out QC, but there is a clearly striking patterns here where reads and genes go down and mitochondrial percentage goes up in the later time points, either indicating biology or technical problems. Either way, attention should be given to this pattern and I did not find it discussed in the manuscript. Authors might want to consider if there is merit in making a more conservative threshold analysis (e.g. only including cells with < 10% mitochondrial percentage) and also annotating this if there is some chance that these patterns might be technical rather than biological in nature.

Minor comments

- The abstract could now better reflect the additional work the authors have done in revisions

Line 713 – please state more clearly how you used the iSeq100 data to determine sequencing depth

Line 715 – please give typical reads per cell style numbers rather than “deep coverage”.

Referee #3

(Remarks to the Author)

I remain convinced that this revised version of the manuscript by Yan and co-authors is highly relevant because it paints the most detailed picture yet of *P. falciparum* development in its vector. To my knowledge, few, if any, biological systems have so far allowed pathogen-host interactions to be dissected by single cell transcriptomics at the level of entire organs in vivo. In my view this adds to the broader appeal of this study.

The combined single cell analysis of parasite and vector provides an important resource and also generates various new findings which the authors explain well in their response to reviewer 1. Among these is the requirement of the PfSIP2 transcription factor for the formation of hepatic stages. Reviewer 1 finds the inclusion of these data “awkward”, but I believe the authors are correct to interpret the phenotype as a consequence of an earlier developmental role of the transcription factor, either in the oocyst where the gene is transcribed, or in the sporozoite where the protein may persist. Other interesting starting points for more mechanistic work include the association of traversing parasites with progenitor cells and the proximity of later oocysts to muscle cells.

Reviewer 1 raises as a major concern that findings from this study may be limited to the specific parasite-vector combination initially investigated by the authors. While it would be unreasonable, in my view, to repeat the entire study with different *P. falciparum* isolates and mosquito species, the authors address this concern adequately by showing that conclusions derived from their transcriptomic analysis are generalisable. They provide new evidence that the transmission blocking compounds aimed at parasite metabolism are active in a different mosquito species and with an artemisinin-resistant *P. falciparum* field isolate. Reassuringly they also now validate their confocal analysis further, confirming an association of traversing ookinetes with progenitor cells not only with an additional field isolate of *P. falciparum* (P5) but also with a recently colonized *An. coluzzii* mosquito. These new data strengthen the manuscript significantly by demonstrating that findings are not specific to a particular combination of parasite and vector.

New data in response to reviewer 2 include critical additional quality control of the mosquito single cell data and critical technical details in the results and methods sections, which are important improvements to the manuscript. My comments have been addressed adequately.

In summary, I find that the revised and improved manuscript responds adequately to all points raised. I continue to be enthusiastic about this work and would be glad to see it in Nature.

Minor point:

Page 9. “Dimensional reduction” should be “dimensionality reduction”.

Version 2:

Reviewer comments:

Referee #2

(Remarks to the Author)

I am satisfied with the responses and the minor additions to the manuscript. There is no need to include 25% cut off analysis in the manuscript -- if there were to be another inclusion, I think it should be much lower (e.g. remove all cells with > 10% mitochondrial fraction and compare to > 30% removed). Irrespective of what such a comparison shows, I think this would be a sensible addition to the supplement given the track record now being established that 25%/30% is ok simply based on citing other papers that used it rather than by showcasing the results when stricter thresholds are applied. If the high mitochondrial fraction is biological, it would also be relevant to ask why some cells have lower fractions than others within the same tissue and if that is cell-type related. Essentially, I think it is a difficult situation where the “truth” is not known and so offering up a more conservative version may be helpful for the field in general so that going forward, 30% is not always considered to be the correct arbitrary threshold. I leave it up to authors to decide if this would be useful in their opinion. From my perspective, it would be useful but I don't think it is required for acceptance/publication.

Point-by-point response to reviewers

Referee #1 (Remarks to the Author):

In Yan et al., the authors assert that they have advanced our knowledge (a "momentous step", line 304) of malaria parasite transitions at the single cell RNA and cellular scales in Anopheles gambiae. If we accept the description of gaps in knowledge in the background, which lacks acknowledgement of previous work in some uncomfortable ways, there is an argument here. However, the data presented are more accurately contextualized as bridging some gaps in previous work that should be better acknowledged.

We apologize for making you uncomfortable with the way we acknowledged previous work, it surely was not our intention to neglect previous literature relevant to this manuscript. Given you did not indicate which references were omitted, we expanded our initial literature searches by conducting several rounds of systematic PubMed searches using terms such as "ookinete invasion", "single-cell RNA sequencing/scRNAseq" and "ookinete midgut interaction". These searches led to over 300 relevant scientific papers concerning previous transcriptomic work and earlier work on parasite traversal of the mosquito midgut epithelium, as well as on midgut physiology. Within the boundaries of a strict citation limitation, we added seven additional references in the main text and 21 in the extended material, listed below.

Relevant additional references in main text:

- 6 Akinosoglou, K. A. et al. Characterization of Plasmodium developmental transcriptomes in Anopheles gambiae midgut reveals novel regulators of malaria transmission. Cell Microbiol 17, 254-268 (2015). <https://doi.org/10.1111/cmi.12363>
- 7 Ukegbu, C. V. et al. Identification of genes required for Plasmodium gametocyte-to-sporozoite development in the mosquito vector. Cell Host Microbe 31, 1539-1551 e1536 (2023). <https://doi.org/10.1016/j.chom.2023.08.010>
- 8 Witmer, K. et al. Using scRNA-seq to identify transcriptional variation in the malaria parasite ookinete stage. Front Cell Infect Microbiol 11, 604129 (2021). <https://doi.org/10.3389/fcimb.2021.604129>
- 17 Abraham, E. G. et al. Analysis of the Plasmodium and Anopheles transcriptional repertoire during ookinete development and midgut invasion. J Biol Chem 279, 5573-5580 (2004). <https://doi.org/10.1074/jbc.M307582200>
- 18 Srinivasan, P. et al. Analysis of the Plasmodium and Anopheles transcriptomes during oocyst differentiation. J Biol Chem 279, 5581-5587 (2004). <https://doi.org/10.1074/jbc.M307587200>
- 19 Boonkaew, T. et al. Transcriptome analysis of Anopheles dirus and Plasmodium vivax at ookinete and oocyst stages. Acta Trop 207, 105502 (2020). <https://doi.org/10.1016/j.actatropica.2020.105502>
- 20 Preira, C. M. F. et al. A time point proteomic analysis reveals protein dynamics of Plasmodium oocysts. Mol Cell Proteomics 23, 100736 (2024). <https://doi.org/10.1016/j.mcpro.2024.100736>

Additional references in the extended material:

- 65 Wang, S. et al. A cell atlas of the adult female Aedes aegypti midgut revealed by single-cell RNA sequencing. Sci Data 11, 587 (2024). <https://doi.org/10.1038/s41597-024-03432-8>
- 66 Germain, P. L., Lun, A., Garcia Meixide, C., Macnair, W. & Robinson, M. D. Doublet identification in single-cell sequencing data using scDbIFinder. F1000Res 10, 979 (2021). <https://doi.org/10.12688/f1000research.73600.2>
- 67 Hixson, B. et al. A transcriptomic atlas of Aedes aegypti reveals detailed functional organization of major body parts and gut regional specializations in sugar-fed and blood-fed adult females. Elife 11 (2022). <https://doi.org/10.7554/eLife.76132>
- 80 Vizioli, J. et al. Gambicin: a novel immune responsive antimicrobial peptide from the malaria vector Anopheles gambiae. Proc Natl Acad Sci U S A 98, 12630-12635 (2001). <https://doi.org/10.1073/pnas.221466798>
- 81 Warr, E., Aguilar, R., Dong, Y., Mahairaki, V. & Dimopoulos, G. Spatial and sex-specific dissection of the Anopheles gambiae midgut transcriptome. BMC Genomics 8, 37 (2007). <https://doi.org/10.1186/1471-2164-8-37>
- 82 Hixson, B., Taracena, M. L. & Buchon, N. Midgut epithelial dynamics are central to mosquitoes' physiology and fitness, and to the transmission of vector-borne disease. Front Cell Infect Microbiol 11, 653156 (2021). <https://doi.org/10.3389/fcimb.2021.653156>
- 83 Hung, R. J., Li, J. S. S., Liu, Y. & Perrimon, N. Defining cell types and lineage in the Drosophila midgut using single cell transcriptomics. Curr Opin Insect Sci 47, 12-17 (2021). <https://doi.org/10.1016/j.cois.2021.02.008>
- 84 Zhang, P. & Edgar, B. A. Insect Gut Regeneration. Cold Spring Harb Perspect Biol 14 (2022). <https://doi.org/10.1101/cshperspect.a040915>

- 85 O'Brien, L. E., Soliman, S. S., Li, X. & Bilder, D. Altered modes of stem cell division drive adaptive intestinal growth. *Cell* 147, 603-614 (2011). <https://doi.org/10.1016/j.cell.2011.08.048>
- 86 Joly, A. & Rousset, R. Tissue adaptation to environmental cues by symmetric and asymmetric division modes of intestinal stem cells. *Int J Mol Sci* 21 (2020). <https://doi.org/10.3390/ijms21176362>
- 87 Baton, L. A. & Ranford-Cartwright, L. C. Morphological evidence for proliferative regeneration of the *Anopheles stephensi* midgut epithelium following *Plasmodium falciparum* ookinete invasion. *J Invertebr Pathol* 96, 244-254 (2007). <https://doi.org/10.1016/j.jip.2007.05.005>
- 88 Taracena-Agarwal, M. L. et al. The midgut epithelium of mosquitoes adjusts cell proliferation and endoreplication to respond to physiological challenges. *BMC Biol* 22, 22 (2024). <https://doi.org/10.1186/s12915-023-01769-x>
- 89 Janeh, M., Osman, D. & Kambris, Z. Damage-induced cell regeneration in the midgut of *Aedes albopictus* mosquitoes. *Sci Rep* 7, 44594 (2017). <https://doi.org/10.1038/srep44594>
- 90 Zeng, X. & Hou, S. X. Enteroendocrine cells are generated from stem cells through a distinct progenitor in the adult *Drosophila* posterior midgut. *Development* 142, 644-653 (2015). <https://doi.org/10.1242/dev.113357>
- 91 Baia-da-Silva, D. C. et al. Microanatomy of the American malaria vector *Anopheles aquasalis* (Diptera: Culicidae: Anophelinae) midgut: ultrastructural and histochemical observations. *J Med Entomol* 56, 1636-1649 (2019). <https://doi.org/10.1093/jme/tjz114>
- 92 Brown, M. R., Raikhel, A. S. & Lea, A. O. Ultrastructure of midgut endocrine cells in the adult mosquito, *Aedes aegypti*. *Tissue Cell* 17, 709-721 (1985). [https://doi.org/10.1016/0040-8166\(85\)90006-0](https://doi.org/10.1016/0040-8166(85)90006-0)
- 93 Dou, X., Chen, K., Brown, M. R. & Strand, M. R. Reciprocal interactions between neuropeptide F and RYamide regulate host attraction in the mosquito *Aedes aegypti*. *Proc Natl Acad Sci U S A* 121, e2408072121 (2024). <https://doi.org/10.1073/pnas.2408072121>
- 94 Hecker, H. Structure and function of midgut epithelial cells in Culicidae mosquitoes (Insecta, Diptera). *Cell Tissue Res* 184, 321-341 (1977). <https://doi.org/10.1007/BF00219894>
- 95 Park, S. S. & Shahabuddin, M. Structural organization of posterior midgut muscles in mosquitoes, *Aedes aegypti* and *Anopheles gambiae*. *J Struct Biol* 129, 30-37 (2000). <https://doi.org/10.1006/jsbi.1999.4208>
- 96 Korzelius, J. et al. The WT1-like transcription factor Klumpfuss maintains lineage commitment of enterocyte progenitors in the *Drosophila* intestine. *Nat Commun* 10, 4123 (2019). <https://doi.org/10.1038/s41467-019-12003-0>
- 97 Resende, L. P., Truong, M. E., Gomez, A. & Jones, D. L. Intestinal stem cell ablation reveals differential requirements for survival in response to chemical challenge. *Dev Biol* 424, 10-17 (2017). <https://doi.org/10.1016/j.ydbio.2017.01.004>

Further, quite a few findings are largely unsurprising, with upregulated genes for cytoskeletal transitions during midgut invasion and increased aerobic respiration with oocyst growth, for example, with only a few novel findings (e.g., PfSIP2-dependent invasion of hepatocytes).

We agree that the upregulation of cytoskeletal and translation-related genes is expected and believe that their identification further supports the quality of our scRNA-seq data. We clarified this in our writing, for example, by writing “expected differences” when describing the identification of cytoskeletal genes. However, several other findings were unexpected and as pointed out by another reviewer, completely novel. Overall, our study, rather than merely confirming previously known parasite biology, crucially identifies novel processes involved in poorly characterized parasite transitions and functionally validates key genes using chemical and genetic approaches. Key novel findings include:

(1) We identified the transcriptome of poorly characterized parasite transition stages, such as the ookinete-oocyst transition (Took) and sporozoite segmentation. These stages were captured due to our carefully planned experimental design—which spanned across different mosquito metabolic states and several time points—and in depth scRNA-seq analysis. A prominent result is the detection of significant *PfEMP1* expression during the Took stage (second branch of our co-expression analysis, Fig. 1g), which led us to hypothesize their roles in anchoring oocysts between midgut cells and basal lamina – an idea that Reviewer 3 described as “*provocative*”. Additionally, we show that sporozoite segmentation is regulated at least in part by the same machinery that regulates merozoite formation in asexual blood stages, despite the sheer difference in the magnitude of these processes (tens of merozoites compared to thousands of sporozoites).

(2) With this wealth of data in hand, it is now possible to identify targets involved in parasite survival and growth. Indeed, we validated these targets using chemical and genetic approaches. We demonstrated that *PfATP4* and *PfLRS* are crucial for oocyst growth using antimalarial drugs, and following your suggestion, we further confirmed their essential roles across *P. falciparum* field isolates, including an artemisinin-resistant strain, and in both African and Asian *Anopheles* species. We went further to demonstrate that *PfSIP2*, a transcription factor expressed during early sporozoite segmentation, is essential for hepatocyte invasion. For this analysis, we adapted a conditional gene knockdown system, which, to our knowledge, is the first application of this system at this stage in human malaria parasites, “*pushing beyond the limits of current technologies*,” as mentioned by another reviewer. These findings underscore the translational potential of our data for developing novel transmission-blocking strategies.

(3) Leveraging scRNA-seq of both mosquito cells and parasites, we were able to unveil previously unappreciated parasite-vector interactions. We were surprised to find that parasites associate strongly with mosquito progenitor cells as they traverse the midgut. This is an intriguing and novel finding that opens new research questions that will undoubtedly be addressed by the community in future studies. Also intriguing is the finding that, contrary to what observed in rodent *P. berghei* parasites, *P. falciparum* oocysts closely interact with muscle cells, potentially to anchor themselves to the midgut epithelium without inflicting damage.

For all these reasons, we believe this is a transformative study that will be of deep value to the entire parasitology and entomology communities and beyond, and we are confident we have further demonstrated its relevance by expanding our findings to diverse parasite-mosquito populations.

As an aside, the latter is awkwardly included, given the focus and title on parasite development in the mosquito host.

While we appreciate your perspective, we respectfully disagree. *PfSIP2* is a transcription factor expressed exclusively in the segmenting oocyst stage, while it is not detectable in salivary gland sporozoites (PMID: 34992211; 34045457; 31439762). Although its function is relevant during hepatocyte invasion, our current evidence suggests this is due to *PfSIP2* regulation of gene expression in oocysts. Importantly, we validate this gene as a key determinant of the parasite transmission cycle, with potential for the identification of downstream transmission-blocking factors.

Drugs (CIP, MMV5) are tested at very high single concentrations (100uM); the choice of this dose is not explained, and effects should be presented over a range of doses for this first use of these drugs in mosquitoes.

We apologize for our limited explanation. We selected the 100 μ M dose because this concentration was used in a previous study based on antimalarial exposure in mosquito via sugar feeding (PMID: 35687594). Besides better explaining our rationale in the methods, we followed your suggestion and tested a range of concentrations. We have now included a dose-response analysis for cipagarmin (CIP). These additional data, presented in Fig. 2d and Extended Data Fig. 4c,d, demonstrates that reducing CIP concentration even 100-fold (from 100 μ M to 1 μ M) still significantly reduces oocyst growth, as indicated by the smaller oocyst size and a large decrease in sporozoite intensity in salivary glands. We did not perform similar experiments with MMV670325 (targeting LRS) due to the high costs of this drug and its limited commercial availability (obtaining just 5 mg required several months) and each sugar feeding experiment consumes more than 1.3 mg. Nevertheless, we tested activity of this compound in an additional mosquito-parasite combination, confirming and expanding our original results.

The authors point out in the discussion that P. falciparum is distinct in its biology from P. berghei. While this is obviously true, the work presented is focused on a single P. falciparum strain in a single strain of A. gambiae. What we really need to know is how variable such findings are across different P. falciparum strains in different strains of A. gambiae, perhaps even in A. coluzzii and in A. stephensi. Such comparisons would reveal how fundamental and applicable these findings are that the authors state are already plausible targets for transmission blocking. Without data on other falciparum strain/mosquito strain and species combinations, we lack a critical understanding as to whether these findings are unique to this parasite-mosquito combination or truly biologically fundamental for publication at the hoped for level.

We thank you for this valuable suggestion and have performed several additional experiments, detailed below, which we believe have made our data stronger and our conclusion more widely applicable.

First, we used our compounds in a completely different mosquito strain-parasite isolate combination. Specifically, we infected *An. stephensi* mosquitoes with the artemisinin-resistant Cambodian field isolate ART29 prior to exposing them to CIP and MMV670325 (Fig. 2e, Extended Data Fig. 4e). The results show comparable or even stronger phenotypes than those observed in our initial *P. falciparum* NF54–*An. gambiae* combination. Indeed, both drugs severely reduced oocyst growth and completely abolished sporozoites in salivary glands at 14d pi (the higher potency of MMV670325 compared to the original results may be due to the fresh batch of the compound that we purchased for these revisions).

Second, we tested CIP on another *P. falciparum* field isolate (isolate P5), this time from Burkina Faso, in *An. gambiae*. Again, we observed severe oocyst growth inhibition and no sporozoites in the salivary glands (Extended Fig. 4f). These are parasites that are currently circulating in African children, therefore highly relevant to ongoing malaria transmission and burden.

Of note, while we inferred that your comment primarily referred to drug-based transmission-blocking experiments, we nevertheless determined whether the observed association of parasites with intestinal progenitor cells during midgut traversal is conserved in other mosquito-parasite combinations. Using confocal microscopy, we quantified the parasite association with intestinal progenitor cells in the Burkina Faso isolate P5 in *An. gambiae*, and in a highly transmissible laboratory NF54_H isolate in *An. coluzzii* (Extended data Fig. 7g, h). In both cases, the association was comparable to the one originally observed in the NF54–*An. gambiae* infection model, with 10.9% of P5 isolates and 8.71% of NF54_H interacting with an ISC, respectively.

We believe these additional experimental validations have strengthened our manuscript, as they demonstrate that our findings are applicable to different *Anopheles* species and parasite isolates, including field isolates from Asia and Africa and a drug-resistant strain.

Beyond the authors surprising assertion that "How surviving ookinetes interact with midgut cells and form oocysts is unknown," which is more accurately described as "poorly" known, there are other examples of inaccurate statements.

We have revised this and other statements for accuracy. The sentence now reads “*How surviving ookinetes interact with midgut cells and form oocysts is poorly known, yet these steps are essential to initiate a remarkable growth process culminating in the production of thousands of infectious sporozoites.*”

We also revised the description of the ookinete-oocyst transition from “uncharacterized process” to a “an as yet poorly characterized process” when discussing the co-expressed genes in the second branch of the gene cluster analysis.

In lines 325-327, the authors write that "Recent studies suggest that..." with citations 4, 12 and 42. Citation 12 is from 2005, citation 4 is from 2007 and citation 42 is the only recent publication from 2023. This is picky, yes of course, but it fits unfortunately with the impression of there being more than a handful of over-statements and inaccurate statements in the text.

We removed the word 'recent' and have now updated the sentence as follows: "The best supported model for midgut crossing to date suggests that ookinetes initially invade an epithelial cell, then move laterally to adjacent cells before reaching the basal side, often leading to the extrusion of the invaded cells."

*Of note from their mosquito biology data, the authors need to clearly cite previously published, definitive work that has confirmed the existence and functional biology of distinct *A. gambiae* midgut cell types (e.g., ISCs/EBs, pECs, aECs, etc). These descriptions have become established through copied inferences from a few papers rather than a citation of works describing functionally validated, definitive cell types in *A. gambiae* or anophelines in general. I know of no such work that has definitively and functionally identified these midgut cell types in *A. gambiae*, but hopefully the authors can clarify this in a revision.*

This is an interesting point, and as you point out, no studies so far have functionally confirmed the cell type annotations in *Anopheles* midguts. The only exception is progenitor cells, which are capable of replication and have been studied in previous work (PMID: 17575986, 38281940). The inferences made here are, however, strongly substantiated by the transcription profile of the cell clusters and by several observational studies describing cell populations in mosquito midguts (PMID: 15648689, 31321415, 4060146). Enterocytes are found to express enzymes related to digestion and transporters for nutrient absorption – hallmarks of this cell type. Enteroendocrine cells are found to express well characterized markers, such as *prospero*, which was initially characterized in *Drosophila melanogaster* and is conserved in *Aedes aegypti* (PMID: 33188922, 38839790, 31915294).

Following this comment as well as comments from another reviewer, we have now provided a more detailed description of our cell cluster annotation in the text and methods, as well as more details of the stepwise analysis of the mosquito data in the supplementary note, ensuring relevant work is appropriately cited and more in-depth information is given to readers interested in mosquito midgut physiology.

Overall, I would judge this work as a very good, incremental addition to the literature for a more focused audience, but there is also work to be done to address over-interpretations and inaccurate statements, an uncomfortable lack of acknowledged previous work (even with the citation limitation this is entirely possible), and some writing challenges that detract from readability.

We hope we have addressed all of your concerns and that our additional data will reassure you of the importance of this study. We apologize for the 'writing challenges' you encountered, and in the absence of specific details we have endeavored to improve the overall readability of this revised version of the manuscript.

*In addition, there needs to be at least partial confirmation of major observations using multiple combinations of *falciparum* strains and *A. gambiae* strains/other major *Anopheles* vector species to deliver us to an understanding of biology that is truly fundamental to *P. falciparum* invasion and development within *Anopheles* spp.*

We appreciate this comment and have addressed it above.

Referee #2 (Remarks to the Author):

This is comprehensive study that uses scRNAseq well, identifying patterns, and then testing the potential importance of some genes of interest in human malaria parasites. The work targeting some follow up genes is solid and convincing. A bit more detail on whether other genes were also tested and found to have no effect, as well as how these genes in particular were decided on for functional assays would be welcome.

Thank you for this positive feedback. Following these comments, we tested two additional candidate genes and found their inhibition yielded intermediate phenotypes in parasite transmission. Specifically, we tested the roles of elongation factor 2 (*PfeEF2*) and the lactate transporter *PfFNT* using MMV643121 and MMV007839, respectively. Inhibition of *PfeEF2* significantly reduced oocyst growth at 7d pi, but this reduction was modest and did not result in a significant effect on sporozoite intensity or prevalence at 14d pi. In contrast, inhibition of *PfFNT* did not impact oocyst size at day 7 post infection, but it led to a significant reduction in the number of sporozoites in salivary glands at 14d pi (Extended Data Fig. 4g, h), suggesting a role for this lactate transporter in sporozoite segmentation or salivary gland invasion. Of note, although our observation that *PfeEF2* inhibition does not decrease sporozoite numbers might suggest that this gene is not essential for parasite development in these stages, this result may also be due to rapid metabolism of the drug in the mosquito, as discussed in the revised manuscript. Conditional knock-outs would definitively prove essentiality, but we believe this vast amount of additional work is beyond the scope of the current manuscript.

To explain our rationale for candidate selection, this was based on two criteria. We prioritized genes from different gene clusters to validate our scRNA-seq results and gene cluster analysis. The initial candidates, *PfATP4* and *PfLRS*, belonged to cluster 5 and 1, respectively. Therefore, in this new analysis we chose candidates from other clusters: *PfeEF2* (cluster 12) and *PfFNT* (cluster 7). Moreover, we selected candidate genes for which potent inhibitors are commercially available. The new compounds have been validated for their target-compound interactions and exhibit sub-micromolar activity against *P. falciparum* asexual blood stages (PMID: 26085270; 34499638). We have better described that rationale behind candidate selection in the revised manuscript.

The authors' study of parasite interactions with mosquito cells gives nice resolution about interactions at these critical time points. While the results look fairly clean, the mosquito data are not shown to have been fully QC'd and this should be remedied to bring the manuscript to the standard of a Nature paper.

Thank you for pointing out this issue. We had indeed performed QC analysis of both mosquito and parasite cells but regrettably neglected to present the mosquito data in a comprehensive form. Following this and other comments, we checked and repeated all steps of our initial QC and realized that we had used an incorrect list of mitochondrial genes for the mosquito analysis. We have corrected this issue by downloading (and confirming mitochondrial genome location of) the latest information from vectorbase.org. We then performed a more stringent QC analysis using a 30% threshold for mitochondrial percentage (based on other scRNA-seq work in insects, PMID: 31915294, 38839790, 39869679) and 0.7 cutoff for cell complexity. These additional steps allowed us to remove additional lower quality cells potentially compromised during the isolation process (mitochondrial percentage) or cells under stress for which only a few genes are detected despite large amount of mRNA (cell complexity). In the methods, we have now separated the parasite and mosquito analyses for clarity, and all QC parameters are presented in Supplementary Table 1.

Please note that although the UMAPs have necessarily changed after this new QC, the only real difference relative to our initial analysis is that the visceral muscle cells are now strongly reduced in number due to their high mitochondrial content. Therefore, the association of late oocysts (7 day post infection) with muscle cells is no longer significant in the scRNA-seq analysis and has been removed from this revised manuscript. Importantly, however, we maintain the confocal microscopy observations of this strong association at the same time point, where we find muscle fibers firmly wrapped around oocysts, potentially providing parasites with a way to ‘anchor’ to the midgut epithelium. To validate its biological relevance, we expanded this observation to other mosquito-parasite combinations, including using *P. falciparum* field isolates (Extended Data Fig. 8 a-c). In all cases, we found the same strong association between muscle fibers and oocysts, suggesting an evolutionarily conserved adaptation between mosquito hosts and parasites.

Appreciating that the details must be reserved for the Methods section, it would be beneficial to the reader to add in a sentence or two to the main text discuss how you isolated parasites and how you are sure in this process that the mosquito cells remained sufficiently intact to warrant joint scRNAseq analysis. Similarly, how you did QC and how many mosquito cells were assayed deserve a brief mention in the main text.

We agree and have provided core information of the single-cell isolation protocol in the main text, and expanded it substantially in the methods section. To briefly clarify the protocol here: we rapidly dissected *P. falciparum*-infected midguts in cold PBS to preserve cell viability. We then dissociated the midguts using a cocktail of 1 mg/ml collagenase IV and elastase at 30°C, shaking at 300 rpm for 30 mins, pipetting up and down every 10 mins. Cell clumps – mostly containing midgut cells – were removed by sequential filtering through 70 µm, 40 µm, and 20 µm cell strainers, allowing isolation of single midgut cells and enrichment of parasites (see response to a later comment for more detail). Mosquito cell viability was assessed using trypan blue staining for each sample, which selectively stains dead cells due to their compromised membranes. By counting trypan blue-positive and -negative cells using a hemocytometer prior to loading onto the 10X chip, we determined that over 93% of mosquito cells were viable. Additionally, we did not detect any trypan blue-positive parasites at day 4 and 7 post infection – the only time points when parasites are visible on hemocytometers due to their larger size – suggesting that most parasites were also viable.

We have revised the main text to briefly explain the isolation process as follows: “*Briefly, infected midguts were partially digested with collagenase IV and elastase, then filtered through a series of cell strainers to remove large mosquito cell clumps. The resulting single-cell suspensions contained both midgut cells and parasites, with cell viability averaging over 93% as determined by trypan blue staining.*”

Based on your suggestion, we have now provided significant additional information in the main text and methods on the QC of parasites and mosquito cells. In the main text, the parasite QC now reads as follow: “*After removing low-quality parasites with low transcript and gene counts and high mitochondrial percentage, we successfully profiled 3,495 parasites ...*”, while the mosquito QC now reads: “*After quality control using read and gene counts, mitochondrial read percentages, and complexity score (Extended Data Fig. 6a, Supplementary Table 1), we successfully profiled 55,789 high-quality midgut cells.*”

In Methods, please be clearer on how many 10x runs were completed and how you addressed batch effects if each treatment/time point was on a separate run. Provide details on cells loaded vs recovered for each run. Also, there is no mention of doublet removal – how did you address doublets?

We have now expanded the methods section and Supplementary Table 1. In total, we analyzed 32 samples across four time points, two metabolic conditions (dsGFP and dsEcR), and four biological replicates. These samples were processed using 16 10X Chromium chips (so 16 runs), with dsGFP and dsEcR samples from the same time point loaded into two separate wells of the same chip. We set a target cell recovery of 4,000 cells per sample and obtained – after QC – an average of 2,962 high-quality midgut cells per sample. This information, along with QC parameters, is now included in Supplementary Table 1. We have also expanded this section of the method to clarify our protocol. The revised text now reads *“Single-cell libraries were prepared following the Chromium Next GEM Single Cell 3’ Reagents Kit v3.1 (Dual Index) User Guide (RevC). Briefly, single-cell suspensions were diluted to target a recovery of 4,000 cells per sample, then mixed with reverse transcription reagents and barcoded gel beads. The mixtures were loaded into wells containing partitioning oil on a 10X Chromium chip to generate single-cell emulsions. Samples from dsGFP and dsEcR mosquitoes collected at the same time point were loaded into two separate wells on the same chip. In total, 16 chips were used across four time points and four biological replicates.”*

We integrated each sample using RPCA analysis to correct for batch effects. This is now explicitly stated in the methods sections for both the parasites and mosquitoes. The updated parasite section now reads: *“Integration to correct for batch effect and differences in single cell methodology was conducted using reciprocal PCA (RPCA), which is the preferred approach when minimal overlap between datasets is expected”*. The mosquito method now reads: *“Integration was conducted using RPCA with the highest k.weight (64) to correct batch effect while avoiding overcorrecting for biological variation between time points and treatments.”*

Doublet removal was performed in both parasite and mosquito single cell analyses. Parasite doublets were removed using Scrublet in Python while mosquito doublets were removed in R using scDbfFinder. While both algorithms show strong performance, scDbfFinder performs particularly well in large complex datasets and was preferred to Scrublet despite the longer run time. We have clarified this further in the methods, and the detailed stepwise analyses are available in the respective code.

In Methods, please expand on the work you did to confirm that the dissociation protocol led to viable parasite and mosquito cells. There is an early reference to “meticulously optimizing the protocol” and to see Methods for more details but these details are not provided in Methods. It is therefore unclear how you enriched for parasites or what optimizations were carried out. Some further expansion is warranted to showcase this optimization and bring more credibility to the protocol that results in good parasite AND mosquito cells (noting as below, this should also be published as a protocol). Did you do any staining for live/dead cells? If the protocol you use has been already shown to lead to viable epithelial tissue single cell suspensions in other studies, please cite the work that showed that. In Methods, you state that mitochondrial percentage per cell was used in quality control only for Plasmodium. This should be done for Anopheles as well. Why wasn’t it done? Epithelial tissue is not straightforward to dissociate (hence many studies resort to single nuclei RNAseq) and the protocol, if it is demonstrated to lead to high quality mosquito gut cells, should be published in detail.

We realize we did not clearly present the optimization steps behind our protocol to ensure isolation of high-quality parasite and mosquito midgut cells. We now have significantly expanded the single-cell isolation section of the methods to describe these steps and clarify the enrichment strategy. We wrote “meticulously optimizing the protocol” because no established method exists for isolating both parasites and mosquito cells simultaneously. Our study is the result of over six months of trials through 22 experimental iterations, as described below.

We began with a protocol that uses collagenase IV and elastase, based on *Drosophila* and *P. berghei* scRNA-seq studies (PMID: 31915294; 31439762), and systematically tested variables such as digestion time, temperature, tissue preparation (intact or minced gut), types of pipette tips. Each attempt was assessed using trypan blue staining to quantify live, single midgut cells as described above. Although we recovered about 400 live, single cells per mosquito midgut, many cells remained in clumps. Given the low number of parasites (~20-30) relative to mosquito cells (~5,000) in a gut, rather than further optimizing for complete dissociation of the tissue, we chose to remove cell clumps – composed mostly of midgut cells – to enrich the parasite-to-mosquito cell ratio. Therefore, we tested multiple cell strainers and found that sequential filtration through 70, 40, and 20 µm strainers yielded the best single cell suspension with minimal cell clumps. We evaluated parasite enrichment using 4d pi and 7d pi samples in trypan blue staining solution, where oocysts could be distinguished from midgut cells based on morphological differences. In the final four replicates that were used for the analysis, after QC, we recovered 55,789 high-quality mosquito cells and 3,495 high-quality parasite cells, yielding an approximate 16:1 mosquito-to-parasite ratio. Based on an estimated 5,000 midgut cells and 30 oocysts per gut – an expected ratio of 166:1 – our protocol represents a 10-fold enrichment of parasites.

While in the text we briefly mentioned not using FACS or MACS for enrichment, we want to clarify that we tested both approaches extensively. Due to the lack of fluorescent parasite lines, we used DNA staining and parasite membrane Pfs25 labeling to identify (1) parasites, (2) midgut cells, and (3) parasite-associated midgut cells for FACS. However, we were unable to achieve clear separation of these populations using these strategies, primarily due to signal bleed-through. Although MACS enriched young parasites via their iron-containing hemozoin, it increased processing time and reduced cell viability, and yield. Ultimately, we selected the cell strainer method as it preserved the integrity and viability of both parasite and mosquito cells.

We have now consolidated the mitochondrial percentage as part of QC for mosquito cells, as described above.

More information on the Anopheles cells should be presented in the main text before the “parasite association with midgut cells” section. This includes for example how many cells/runs were used and also please provide justification of the merging of Anopheles cells from both control and dsEcR as soon as the Anopheles cells start to be discussed (if indeed that is what was done). It’s otherwise confusing to the reader. Cell type annotations for the Anopheles data are also not explained in enough detail. The manuscript only states “annotated manually by cross-referencing cluster markers from four references”. Please provide more information on how this was done – e.g. what did you trust as enough concordance to call something a particular cell type?

Following your recommendation, as well as comments by other reviewers, we have expanded the first paragraph of the section to focus exclusively on the description of the mosquito data, providing more details on number of cells and runs used in the main text as well as in the method as described above. We also added a supplementary note to expand on the current understanding of insect gut cell types and how this knowledge informed our cluster annotation.

Regarding the analysis of the mosquito data, cells from dsGFP and dsEcR mosquitoes were indeed integrated before clustering and this is now more clearly stated in the updated manuscript as follows: “Cells from each sample were then integrated using RPCA to correct for treatment and batch effects ...” To explain the rationale leading to this decision: similarly to the parasite analysis, we first performed pseudobulk differential expression (DE) analysis of the mosquito dataset comparing the two conditions at each of the four time points. We, however, found no genes were differentially expressed. This validated our decision to merge the data for the single cell analysis and later for interaction with parasites. Post integration and clustering, we performed a second

pseudobulk DE, this time pooling cell types together rather than all cells from each sample. This was due to the fact most midgut cells are enterocytes which could mask signal from other cell types. In this analysis, we found few differentially expressed genes in 3 of the 4 analyses (a summary of the results can be found in Extended Data Fig. 6d with the full list presented in Supplementary Table 6). By contrast, time post bloodmeal had a greater impact on the mosquito transcriptomes (Extended Data Fig. 6e, Supplementary Table 7). These results supported our decision to integrate the ds*GFP* and ds*EcR* samples for the dual-read analysis.

Regarding the cell cluster annotation, we now provide a supplementary note (for space constraints in the main text) with a more detailed stepwise description of the process. In the main text, the process is now concisely described as follows: “*We annotated the 16 resulting clusters based on marker gene expression and GO terms (Fig. 4a, Extended Data Fig. 6b, c, Supplementary Table 5, with a detailed description of each cell type and the rationale behind annotations provided in the Supplementary Note).*” Extended Data Fig. 6 is now updated to show the use of the *An. gambiae* bulk RNA-seq as a reference, and Supplementary Table 5 now includes the functional enrichment analysis performed for each cluster.

To give a little more context here, we used bulk RNA-seq data from the proventriculus and the anterior and posterior midgut of *An. gambiae* (PMID: 35471187) as a reference to infer the general localization of midgut cells in our mosquito UMAP. Next, known gene markers of midgut cells identified in other single cell and single nuclei RNA-seq studies in *Drosophila* and *Aedes* were used to agnostically characterize each cell cluster (PMID: 31915294, 32855340, 33188922, 35239393). Functional enrichment of the cluster markers was then employed to corroborate the annotations, all of which were consistent with the identified cell types. Finally, specific gene markers considered hallmarks of each cell type in *Aedes* and *Drosophila* were used to confirm annotation. Across all 16 clusters, only one cluster did not show a consistent signature of a known cell type and was labelled EE-like as it expresses the main enteroendocrine marker *prospero*.

Minor comments

Main text references to EDF 6c and d should be swapped.

This has been corrected, though please note EDF 6 is now EDF 7 and has changed considerably.

Fig 1b, please put percent of variance explained by each PC.

The figure was updated to indicate explained variance of the first three principal components.

Please comment on the typical size of a 7-day old oocyst and compatibility with any cell size limits of 10x. Presumably this limitation is why some older oocysts were not explored using 10x, but if it was tried and failed, it would be good to report here.

You are correct by presuming we did not explore later time points beyond 7-day post infection. The average diameter of a 7-day old oocyst across the four replicates ranges from 24 μm in ds*GFP* mosquitoes to 26 μm in ds*EcR* mosquitoes, close to the recommended 30 μm upper limit for the 10X platform. We therefore feared that later timepoints would result in oocysts too large to pass through the 10X chip. We have now added the following sentence in the method section: “*We chose not to collect samples beyond 7d pi, as oocysts larger than 30 μm may clog the 10X Chromium chip.*”

The last sentence of the discussion could be removed, it doesn't add a lot of value and this was already clear.

This sentence was removed.

Referee #3 (Remarks to the Author):

*This elegant study presents significant progress by taking our knowledge of parasite-vector interactions in malaria beyond the state of the art. Importantly, it does so not in a tractable rodent model but by looking at the more challenging and most important human malaria parasite, *P. falciparum*. For the first time, this study exploits the power of single cell transcriptomics to analyse the interacting cells of the parasite and vector midgut simultaneously. The combined analysis of parasite and midgut cellular transcriptomes produces an atlas of parasite development in the mosquito that is more highly resolved than previous studies in *P. berghei* or *P. falciparum*, thereby generating an essential new community resource. New clusters of co-regulated parasite genes are identified which can be predicted to enable a deeper understanding of parasite biology. The study additionally characterises cell types of the mosquito midgut at unprecedented depth. Conceptually, the use of single cell transcriptomics to explore the complexity of a parasite vector system is novel and exciting.*

Thank you for your positive comments, we are pleased you found our study conceptually novel and exciting, a significant progress in understanding of vector-parasite interactions and an essential resource for the community.

Examples are provided as proof of principle for how the new data can be exploited in future to develop transmission blocking interventions that target oocyst proteins and for gaining a deeper understanding of parasite biology. Specific biological insights in this manuscript do not provide complete, new mechanisms but provide inspiration and starting points for future research. I am convinced the data will inspire future discoveries. Illustrated starting points include potential targets for blocking sporogony with inhibitors delivered to mosquitoes, the role of a DNA binding protein for the liver stage and the interactions of different parasite stages with host cell types. All these experiments are well performed and push beyond the limits of current technologies, for instance by adapting the conditional gene knockdown system to study oocyst biology and sporozoite infectivity. That oocysts anchor themselves to the midgut using PfEMP1 adhesins and may upregulate muscle cells adhesive genes for the same purpose are provocative ideas that will be interesting to explore in future. The study provides a valuable resource for data, for new ideas, hypotheses and starting points for future work. I consider it to be a landmark study for the field.

The main value of the work is in the overall picture it paints of cell types of vector and parasite and their potential interactions. Individual validation experiments are not overinterpreted, with the exception of the minor points raised below. Deeper mechanistic studies, like visualising or knocking out parasite and muscle cell adhesins, would be possible but could be argued to be beyond the scope of this work, since it would not be necessary to increase my confidence in the utility of the resource and the validity of the overall picture it paint.

Overall, in my opinion the quality of the data is good, their analysis is appropriate, the presentation is clear and the conclusions are supported well by the data.

Again, we are grateful for these comments and flattered that you consider our work a landmark study for the field. We concur that it would be very interesting to follow up on some exciting observations by, for instance, assessing a possible role of PfEMP1 on oocyst adhesion, but due to the complexity of these analyses, we agree that these are better left to future studies.

Minor points:

- Regarding the speculation that upregulated PfEMP1 proteins many anchor oocysts to midgut cells or basal lamina, are these genes uniquely upregulated in oocysts or also (on other*

available data sets) in asexual blood stages? Do they differ regarding their targeting signals, given that blood stages PfEMP1 proteins are targeted to the host cell plasma membrane?

Those are excellent questions that we shared with our collaborator Prof. Kirk Deitsch – a world-renowned expert in PfEMP1 biology. Although for space constraints we cannot add the discussion below in our manuscript, we are pleased to discuss these questions here.

Our scRNA-seq dataset detected upregulation of *var* genes encoding PfEMP1 proteins during the ookinete-to-oocyst transition. These *var* genes are not uniquely expressed in oocysts compared to asexual blood stages, as indicated by current literature. Several *var* genes highly expressed in our dataset – including the most highly expressed gene PF3D7_0809100 – are also expressed during asexual and gametocyte stages (PMID: 28642573; 34045457; 29580379). Interestingly, PF3D7_0809100 is also expressed in late oocysts (10d pi) in *An. stephensi* mosquitoes, as shown in bulk RNA-seq data, and localizes on the surface of sporozoites, playing an important role in liver infection (PMID: 29539423).

Regarding PfEMP1 targeting signals, these PfEMP1 proteins, including PF3D7_0809100, do not differ in any discernable way. Their domain structure consists of canonical DBL and CIDR domains typical of the PfEMP1 family, and they contain the conserved N-terminal segment known to mediate trafficking to the red blood cell surface (PMID: 20438573). Moreover, PF3D7_0809100 is not noticeably conserved in the genomes of different parasite isolates, suggesting it is not specifically required for sporozoites/oocysts. Overall, therefore, there are no clear differences in targeting signals, and how these PfEMP1s are properly trafficked and displayed on the surface of a sporozoite/oocyst remains unknown.

• *Did the authors find progenitor transcriptomes to be influenced by the presence of a parasite?*

We ran several differential expression (DE) analyses to understand if and how mosquito-parasite interactions influence either the parasite or the mosquito transcriptome, but could not identify any significant changes. We restricted our analyses to mosquito and parasite cells from the 36h pi samples, as this is the time point where the association is observed. Specifically, we first compared dual-read parasites to parasites that are not associated with a midgut cell and found no transcriptional differences. We then focused on the specific interaction with ISCs/EBs and again found no significant difference in gene expression compared to other dual-read parasites. These two analyses were performed with different binning to account for variability due to the parasite stage (cluster) or batch (sample), but including either or both variables in the analysis did not lead to different results.

We ran reciprocal analyses with the mosquito data. No genes were found to be differentially expressed between dual-read midgut cells and midgut cells that do not share a barcode with a parasite. Restricting the analysis to progenitor cells—the cluster that contains the most dual-read cells—we again found no differences. Much like the analyses of the parasite data, several iterations of these analyses were performed with different variables (treatment, sample, cell cluster) selected as co-variables, but none yielded significant changes in gene expression.

This lack of positive results is likely due to the limited number of dual-read cells, particularly compared to the large number of mosquito cells, limiting the power in these analyses. We suspect that a tailored experiment specifically focusing on the midgut traversal and sorting parasite-positive midgut cells would be needed to address this question, but this is beyond the scope of the current manuscript. We have nevertheless presented and discussed these comparisons in the revised manuscript as follows: “*In differential expression analyses, no protein coding genes of midgut progenitor cells were affected by the presence of an interacting parasite (FDR = 1), and similarly, parasites interacting with a progenitor cell showed no difference in their transcriptomes*”

relative to those of non-interacting parasites (FDR = 1), although these results may be due to the limited number of dual-read cells.”

Did mosquito cell transcriptomes change with time after feeding or EcR knock down?

We have now included pseudobulk DE analysis in our revised manuscript, grouping mosquito cells according to their cell types and replicates. Only a few genes were differentially expressed between ds*EcR* and ds*GFP* mosquitoes (Extended Data Fig. 6d, Supplementary Table 6), which prevented us from performing functional enrichment analyses. By contrast, midgut transcriptomes changed dramatically with time after blood ingestion. In this case, we chose to focus on control mosquitoes (ds*GFP*), and pseudobulk DE was performed comparing consecutive time points (2d to 36h, 4d to 2d and 7d to 4d post infection) followed by functional enrichment analysis (Extended Data Fig. 6e, Supplementary Table 7).

Methods and results sections were also updated to include the relevant information and discussion on these additions. Updated results are explained as follows: *“Interestingly, comparing dsGFP to dsEcR samples showed limited impact on mosquito transcriptomes, with fewer than 10 genes differentially expressed at each time point (Extended Data Fig. 6d, Supplementary Table 6). On the contrary, time post blood meal strongly affected gene expression across all major cell types (Extended Data Fig. 6e, Supplementary Table 7).”*

• If there is a temporal component, were the transcriptomes of parasite-associated muscle cells compared with un-associated cells from the same sample, or could there be a temporal or batch effect explaining the 24 upregulated genes?

Our initial analysis focused exclusively on visceral muscle (VM) cells 7 days post-infection and compared dual-read VM cells to VM cells that were not associated with a parasite. Several iterations of this analysis were performed to regress out potential confounding variables, such as replicates and/or treatments, but the same 24 genes were consistently found in each analysis.

Following comments by another reviewer, however, we decided to perform more stringent QC analysis of the mosquito data, which unfortunately led to the removal of most visceral muscle cells due to their high mitochondrial content. Consequently, we removed the original parasite-muscle cell dual read analysis from this revised manuscript, although the initial observations of strong physical interactions between muscles and late oocysts were confirmed by confocal microscopy in two other mosquito-parasite combinations, including in field parasites (Extended Data Fig. 8a-c).

To provide more detail in this response letter, we checked and repeated all steps of our initial QC and realized that we had used an incorrect list of mitochondrial genes for the mosquito analysis. We have corrected this issue by downloading (and confirming mitochondrial genome location of) the latest information from vectorbase.org. We then performed a more stringent QC analysis using a 30% threshold for mitochondrial percentage (based on other scRNA-seq work in insects, PMID: 31915294, 38839790, 39869679) and 0.7 cutoff for cell complexity. These additional steps allowed us to remove additional lower quality cells potentially compromised during the isolation process (mitochondrial percentage) or cells under stress for which only a few genes are detected despite large amount of mRNA (cell complexity). In the methods, we have now separated the parasite and mosquito analyses for clarity, and all QC parameters are presented in Supplementary Table 1.

Please note that although the UMAPs have necessarily changed after this new QC, the only real difference relative to our initial analysis is that the visceral muscle cells are now strongly reduced in number due to their high mitochondrial content. Therefore, the association of late oocysts (7 day post infection) with muscle cells is no longer significant in the scRNA-seq analysis and has

been removed from this revised manuscript. Importantly, however, we maintain the confocal microscopy observations of this strong association at the same time point, where we find muscle fibers firmly wrapped around oocysts, potentially providing parasites with a way to ‘anchor’ to the midgut epithelium. To validate its biological relevance, we expanded this observation to other mosquito-parasite combinations, including using *P. falciparum* field isolates (Extended Data Fig. 8 a-c). In all cases, we found the same strong association between muscle fibers and oocysts, suggesting an evolutionarily conserved adaptation between mosquito hosts and parasites.

Given that different size cell strainers were used, could the abundance of Oocyst 4-muscle doublets be the result of selective filtering of the day 7 samples? Are VM cells recovered equally at different time points, i.e. could earlier samples have physically removed muscle cell associated parasites through the strainer with smaller pore size?

We agree that the use of 20 µm strainers could explain the absence of dual-read parasites at 4d pi. While the strainer size might have been a contributing factor, we believe a biological explanation is also involved. In the original dataset, we observed a steady increase in cell number in the VM cluster as time progressed from 36h to 2, 4, and 7d pi, suggesting that the last filtration step is not the only explanation for the higher proportion of VM cells at 7d pi. This issue is however not relevant for the current revision given the new mosquito datasets do not contain many muscle cells.

• *p. 8: They should tone down the conclusion that their data show PfSIP2 expression in late oocysts to be essential for the ability of sporozoites to infect hepatocytes. Although I agree this is likely, it isn't the only possibility. Gene expression in the sporozoite could be important even if below the single cell detection limit, or the protein could act in the sporozoite even though the transcript originates in the schizont. The highest expression level of a gene does not necessarily coincide with the most essential function, and that the sporozoite has some low but important level of transcription is impossible to rule out. Perhaps say “suggest” or include a “most likely”.*

We agree with this observation and have toned down our conclusions, pointing out that other possibilities may exist.

• *p. 9: The suggestion that VM cells respond to the presence of parasites by enhancing their adhesive properties is not the only explanation for the data. The statement should be qualified or spatial/imaging data should be provided that muscle fibres next to cysts express different proteins. Localisation of such proteins by microscopy would be exciting, of course. The single cell data alone may be explained differently. For instance, cells with more adhesins may simply be more likely to remain associated with an oocyst.*

See our response above.

• *Please, explain where the reported effects on splicing can be found in the data.*

We realize our wording may have been misleading. We did not analyze differential splicing events in the data, but rather, we observed that genes associated with helicase activity and RNA splicing are upregulated during the ookinete-oocyst transition. We have revised the manuscript to clarify this issue. The main text now reads: “*The onset of this transition (clusters 13 and 2) was characterized by strong signals of genes encoding for helicase activity and RNA processing, including splicing.*”

Point-by-point response to reviewers

Referee #2 (Remarks to the Author):

The authors have done a highly commendable job responding to all reviews.

Thank you for your positive comment. We are pleased that you found our revisions highly commendable.

I still have some concerns about the quality of the included mosquito single cell data and believe this should be commented on through additional text in the manuscript (not through additional experiments). The authors have now added in more details on mosquito cell QC and annotation. They elect to include mosquito cells up to 30% mitochondrial reads, but this is higher than I have ever heard of and there seems to be a pattern of just citing other papers that have used seemingly high and arbitrary thresholds even among the five papers the authors now cite on line 779. As far as I could tell, only one of the 5 cited papers went as high at 30%. The Fly Cell Atlas paper (which uses nuclei so is going to use lower thresholds) has its liberal setting at 15% and its strict setting at 5% mitochondrial reads. The authors have now included figure S6a that shows the distribution of the mitochondrial read percentage and it is striking that days 4 and 7 actually have the majority of cells showing well above 10% mitochondrial reads while this is not true for 36 hours, where most cells are more “normal”. I would appreciate some attention to this in the manuscript text. Ultimately, these data are what they are and the authors need to make semi-arbitrary cut offs to carry out QC, but there is a clearly striking patterns here where reads and genes go down and mitochondrial percentage goes up in the later time points, either indicating biology or technical problems. Either way, attention should be given to this pattern and I did not find it discussed in the manuscript. Authors might want to consider if there is merit in making a more conservative threshold analysis (e.g. only including cells with < 10% mitochondrial percentage) and also annotating this if there is some chance that these patterns might be technical rather than biological in nature.

We agree with your statement that mitochondrial cutoffs are to an extent arbitrary, which is why we applied a cutoff used by others in mosquito studies. Moreover, we confirmed quality of the cells through other quality control metrics (including viability tests as now reported in the revised manuscript). We detail our reasoning below.

As you noted, some single-cell sequencing studies on insect tissues have used single-nucleus RNA sequencing which is not directly comparable to our study as it is definitely expected to have a lower mitochondrial RNA content threshold than single cell RNA-seq. Of the five previously cited studies as references for our analysis pipeline, three used scRNA-seq as we did, and these are comparable to ours. Indeed, the only study in *Anopheles gambiae* where a mitochondrial reads cutoff was reported used the same 30% threshold as ours (PMID: 32855340, *Science*) and profiled hemocytes, which are already individual cells in the mosquito body cavity and do not require tissue dissociation. The same 30% threshold was also used by the cited study on the *Aedes aegypti* mosquito midgut (PMID: 38839790, *Scientific Data*). These two studies guided our decision on this quality control metric. Two additional scRNA-seq studies in *An. gambiae*—another one of hemocytes (PMID: 34318744, *Elife*) and one of testes (PMID: 37582841, *Communications Biology*)—did not report any mitochondrial read percentage cutoff, so we did not cite them.

The third scRNA-seq study we cited was the gut atlas of *Drosophila melanogaster* (PMID: 31915294, *PNAS*) and this used a very similar cutoff (25%). A study published after our initial submission used this same 25% cutoff in midguts of *Culex tarsalis* mosquitoes (PMID: 39869679,

PLoS Pathogens). We have added it to our citations and have revised the methods section, explaining our rationale for the 30% mitochondrial read percentage cutoff in light of each of the cited studies.

We nevertheless followed your suggestion and re-ran the integration, UMAP, and clustering analyses using a lower (25%) mitochondrial reads threshold, as done in the two scRNA-seq papers mentioned above (*Drosophila* and *Culex*). We have not tried any lower cutoff as this is not done in any other mosquito scRNA-seq studies that we could find in the literature.

As shown below, applying the 25% mitochondrial read percentage cutoff results in only minor changes to the mosquito UMAP and clustering outputs. We still obtain 16 mosquito cell clusters as in Fig. 4a, the only difference being that EE cells are now divided in 3 rather than 4 clusters and pEC cells are divided into 6 rather than 5 clusters compared to the UMAP with 30% mitochondrial cutoff. Importantly, the dual-read analysis at 36h post infection (aimed at identifying mosquito-parasite interactions during midgut traversal) gives the same results as presented in Fig. 4b-d of the manuscript, showing a significant association between parasites and midgut progenitor cells (OR 3.08, $fdr = 0.0315$) and a reduced association between parasites and pECs (OR 0.44, $fdr = 0.0315$) even though the number of dual reads at 36h pi decreased from 107 to 89. At 2d pi we see no significant association between cells, again as previously shown in Fig. 4b using the 30% cutoff.

Below we provide these data in the same format as Fig. 4 of the manuscript.

Parasite interactions with progenitor cells during midgut traversal using 25% mitochondrial read percentage cutoff (compare to Fig. 4 in manuscript)

a, UMAP of *An. gambiae* midgut cells after using 25% mitochondrial read percentage as quality control, showing 16 cell clusters similar to those provided in Fig. 4 of the manuscript using a 30% cutoff. **b**, UMAP plots of *An. gambiae* colored by cell type and overlaid with dual-read cells (grey circles) at 36h pi (left) and 2d pi (right). **c**, UMAP plot of *P. falciparum* from this study overlaid with dual-read parasites (grey circles) at 36h pi after applying the 25% mitochondrial cutoff. **d**, Statistical analysis of the distribution of dual-read parasites in mosquito cell types at 36h pi with the observed vs. expected numbers under the random distribution (left x axis) and resulting odds ratio (\pm 95% confidence interval, right x axis) (Fisher's exact test, *: false discovery rate (FDR) < 0.05).

For the reasons above and the consistent results with 25% cutoff, we maintain our original choice of 30% mitochondrial cutoff is appropriate for our study and in this way the manuscript remains consistent with previous *Anopheles* and *Aedes* scRNA-seq studies. We could nevertheless add the 25% analysis as additional supplementary data, if you consider it necessary for the conclusions of our work. Importantly, we have already confirmed the association between parasites and midgut progenitor cells by confocal microscopy, in the original *Anopheles-Plasmodium* combination, as well as in two additional combinations that include a field-derived *P. falciparum* line. This provides further support to our cutoff choice.

Regarding the differences observed between timepoints, this is a very interesting observation that in our opinion is better explained by several biological factors, given that our isolation and analysis protocol was the same for all samples, and similar viability of cells as assessed by trypan blue was recorded across samples. Mosquitoes at the later timepoints (4 and 7 days post blood meal) are older (8 and 11 days old, compared to 5–6 days old at the first two time points), which likely affects gut physiology, particularly considering the short mosquito lifespan (15-20 days) (PMID: 5854754, *Bull Entomol Res*). Additionally, the time points are not comparable in terms of blood feeding status, as at the later time points blood is completely digested, likely affecting midgut physiology, and polyploidy (PMID: 38281940, *BMC Bio*). Indeed, mitochondrial volume density is seen to increase in several blood feeding insects by electron microscopy after completing blood digestion (PMIDs: 21747, 487393 *Cell Tissue Res*; PMID 11784937, *Mem Inst Oswaldo Cruz*), likely contributing to the increase in mitochondrial read proportion we observed. Finally, the status of the infection also changes as at the later time points, midguts are hosting large, actively growing parasites. While we do not know how these factors may combine, it is reasonable to assume that they influence mitochondrial gene expression. As suggested by you, we added a short description of the observed differences, and the biological factors potentially underlying them, in the methods.

Minor comments

- The abstract could now better reflect the additional work the authors have done in revisions

Thank you for this suggestion. Within the word limitations, we have now added information on the different *Anopheles-Plasmodium* combinations tested.

Line 713 – please state more clearly how you used the iSeq100 data to determine sequencing depth

We initially sequenced our samples on the iSeq100 platform to estimate the proportion of reads mapping to parasites relative to mosquito cells to calculate NovaSeq sequencing depth. This analysis showed that 0.05%, 0.1%, 0.6%, and 1.3% of reads mapped to parasites at 36h, 2d, 4d and 7d pi, respectively. Based on an estimate of ~380 parasites per sample (using a hemocytometer), we calculated ~3,040 parasites per time point (across two metabolic conditions and four replicates). Given the low proportion of parasite-mapped reads, for the final sequencing, we used all lanes of two NovaSeq S4 flow cells to sequence all 32 samples, yielding ~12 billion total reads. This resulted in a predicted 6–156 million reads mapping to parasites from 36h to 7d pi, corresponding to an estimated sequencing depth of 1,974–51,316 reads per parasite from 36h to 7d pi. We have added the rationale for the iSeq run and the sequencing depth estimation in the methods and Supplementary Note (now in SI Guide).

Line 715 – please give typical reads per cell style numbers rather than “deep coverage”.

We have updated the methods to include the expected number of reads per parasite and detailed the calculation in the Supplementary Note (now in SI Guide).

Referee #3 (Remarks to the Author):

*I remain convinced that this revised version of the manuscript by Yan and co-authors is highly relevant because it paints the most detailed picture yet of *P. falciparum* development in its vector. To my knowledge, few, if any, biological systems have so far allowed pathogen-host interactions to be dissected by single cell transcriptomics at the level of entire organs in vivo. In my view this adds to the broader appeal of this study.*

The combined single cell analysis of parasite and vector provides an important resource and also generates various new findings which the authors explain well in their response to reviewer 1. Among these is the requirement of the PfSIP2 transcription factor for the formation of hepatic stages. Reviewer 1 finds the inclusion of these data “awkward”, but I believe the authors are correct to interpret the phenotype as a consequence of an earlier developmental role of the transcription factor, either in the oocyst where the gene is transcribed, or in the sporozoite where the protein may persist. Other interesting starting points for more mechanistic work include the association of traversing parasites with progenitor cells and the proximity of later oocysts to muscle cells.

*Reviewer 1 raises as a major concern that findings from this study may be limited to the specific parasite-vector combination initially investigated by the authors. While it would be unreasonable, in my view, to repeat the entire study with different *P. falciparum* isolates and mosquito species, the authors address this concern adequately by showing that conclusions derived from their transcriptomic analysis are generalisable. They provide new evidence that the transmission blocking compounds aimed at parasite metabolism are active in a different mosquito species and with an artemisinin-resistant *P. falciparum* field isolate. Reassuringly they also now validate their confocal analysis further, confirming an association of traversing ookinetes with progenitor cells not only with an additional field isolate of *P. falciparum* (P5) but also with a recently colonized *An. coluzzii* mosquito. These new data strengthen the manuscript significantly by demonstrating that findings are not specific to a particular combination of parasite and vector.*

New data in response to reviewer 2 include critical additional quality control of the mosquito single cell data and critical technical details in the results and methods sections, which are important improvements to the manuscript. My comments have been addressed adequately.

In summary, I find that the revised and improved manuscript responds adequately to all points raised. I continue to be enthusiastic about this work and would be glad to see it in Nature.

Thank you for your positive comments, we are delighted you are enthusiastic about this work!

Minor point:

Page 9. “Dimensional reduction” should be “dimensionality reduction”.

We have introduced this change.

Point-by-point response to reviewers

Referee #2 (Remarks to the Author):

I am satisfied with the responses and the minor additions to the manuscript. There is no need to include 25% cut off analysis in the manuscript -- if there were to be another inclusion, I think it should be much lower (e.g. remove all cells with > 10% mitochondrial fraction and compare to > 30% removed). Irrespective of what such a comparison shows, I think this would be a sensible addition to the supplement given the track record now being established that 25%/30% is ok simply based on citing other papers that used it rather than by showcasing the results when stricter thresholds are applied. If the high mitochondrial fraction is biological, it would also be relevant to ask why some cells have lower fractions than others within the same tissue and if that is cell-type related. Essentially, I think it is a difficult situation where the "truth" is not known and so offering up a more conservative version may be helpful for the field in general so that going forward, 30% is not always considered to be the correct arbitrary threshold. I leave it up to authors to decide if this would be useful in their opinion. From my perspective, it would be useful but I don't think it is required for acceptance/publication.

Thank you for your comments throughout this review process, which have surely strengthened our work. For our study, we used previously published thresholds, and we believe this was appropriate as we confirmed cell quality through other means (e.g. viability test showing >93% of cells were viable). Additionally, we validated the scRNA-seq finding—the association between parasite and intestinal progenitor cells—via microscopy across different mosquito-parasite species combinations. We therefore decided not to add further analysis to the data.

We agree that moving forward it would be interesting to understand why different cell types have different mitochondrial fractions. For instance, determining the effects of time post blood feed on mitochondrial content is an interesting biological question to better understand blood feeding physiology. Future studies beyond the scope of this work would be needed to specifically address this and other questions related to midgut physiology, given such analyses would most likely require further experimental validation without confounding variables such as infectious status.